

# WAVETRISK-1.0: an adaptive wavelet hydrostatic dynamical core

Nicholas K.-R. Kevlahan[1] and Thomas Dubos[2]

[1]Department of Mathematics and Statistics, McMaster University, Hamilton, Canada
[2]Laboratoire de Météorologie Dynamique, École Polytechnique, Palaiseau, France

**Correspondence:** N. Kevlahan (kevlahan@mcmaster.ca)

**Abstract.** This paper presents the new adaptive dynamical core `wavetrisk`. The fundamental features of the wavelet-based adaptivity were developed for the shallow water equation on the $\beta$-plane in Dubos and Kevlahan (2013) and extended to the icosahedral grid on the sphere in Aechtner et al. (2015). The three-dimensional dynamical core solves the compressible hydrostatic multilayer rotating shallow water equations on a multiscale dynamically adapted grid. The equations are discretized using a Lagrangian vertical coordinate version of `dynamico` introduced in Dubos et al. (2015). The horizontal computational grid is adapted at each time step to ensure a user-specified relative error in either the tendencies or the solution. The Lagrangian vertical grid is remapped using an adaptive Lagrangian-Eulerian (ALE) algorithm onto the initial hybrid $\sigma$ pressure-based coordinates as necessary. The resulting grid is adapted horizontally, but uniform over all vertical layers. Thus, the three-dimensional grid is a set of columns of varying sizes. The code is parallelized by domain decomposition using `mpi` and the variables are stored in a hybrid data structure of dyadic quad trees and patches. A low storage explicit fourth order Runge–Kutta scheme is used for time integration. Validation results are presented for three standard dynamical core test cases: mountain-induced Rossby wave train, baroclinic instability of a jet stream and the Held and Suarez simplified general circulation model. The results confirm good strong parallel scaling and demonstrate that `wavetrisk` can achieve grid compression ratios of several hundred times compared with an equivalent static grid model.

## 1   Introduction

Atmospheric flows are intrinsically non-stationary and multiscale. They are characterized by length scales varying from millimetres to thousands of kilometres and time scales from seconds to decades. Effective climate modelling requires long runs (typically decades) that resolve the dynamically significant scales of motion. However, the smallest significant scales of motion are highly intermittent in space and non-stationary in time. Especially at higher resolutions, the portion of the flow with active small scales is relatively small. Attempting to resolve these important small scales with a uniform grid is necessarily inefficient. Either we need to choose a very fine grid to resolve the smallest scales of interest, which limits simulation times, or we use a coarser grid that filters out these small scales, potentially losing important dynamics. In addition, a uniform resolution does not allow effective control of numerical error. Some regions will be under-resolved and others will be over-resolved. A static com-





putational grid also uses computational resources inefficiently by spending cpu time computing dynamically inactive regions. Ideally, the numerical error should scale predictably with the number of computational elements (e.g. nodes) which should, in turn, be proportional to cpu time. This paper introduces a new wavelet-based adaptive dynamical core, `wavetrisk`, that attempts to achieve these goals.

Behrens (2009) provides a good overview of early developments in adaptive atmospheric modelling, including early work by Skamarock et al. (1989) and previous work using moving grids to model tropical cyclones (Harrison, 1973; Jones, 1977). Since this early work, adaptive mesh refinement (AMR) has found many applications, e.g. fluid dynamics (Popinet and Rickard, 2007) and astrophysics (Mignone et al., 2012; Bryan et al., 2014). In such applications, the main added value of AMR is its ability to accurately capture events that are highly localized in space and time at a relatively low computational cost. Jablonowski

et al. (2009) evaluated a block-structured AMR method for scalar two-dimensional transport on the sphere. Refinement and coarsening levels are constrained so that there is a uniform 2:1 mesh ratio at all fine grid/coarse grid interfaces In a series of test cases they find that the additional resolution helps preserve the shape and amplitude of the transported tracer while saving computing resources in comparison to uniform-grid model runs. Kopera and Giraldo (2014) evaluated the performance of AMR in a discontinuous Galerkin based IMEX model on a planar two-dimensional grid. They found that AMR could provide

up to a 15 times speed-up with minimal overhead. Ferguson et al. (2016) analyzed the performance of an AMR model for the shallow water equations on the cubed sphere. Their model is built using the general purpose Chombo-AMR toolbox of finite difference and finite volume methods for the solution of partial differential equations on block-structured adaptively refined rectangular grids. They test the model for one or two levels of refinement with refinement ratios of $\times 2$, $\times 4$ and $\times 8$ between each level. In comparison, `wavetrisk` currently uses a fixed refinement ratio of $\times 2$, but has been tested for up to six levels of

refinement. Ferguson et al. (2016) conclude that AMR can be effective provided that a sufficiently fine coarsest grid is selected.

    Beyond its ability to accurately capture events that are quite localized in space and time at a low cost, several additional properties are required for AMR to be an attractive option for atmospheric applications. Firstly, AMR must demonstrate its ability to accurately simulate complex three-dimensional flows, where a large number of important features such as cyclones and waves continuously appear, move and disappear. `Wavetrisk` has already demonstrated this ability for (two-dimensional)

shallow-water flows. To the best of our knowledge, no previous work has developed and evaluated AMR for complex three-dimensional atmospheric flows. Furthermore, especially for applications to climate modelling, the robustness of the method over long time scales and its ability to capture accurate statistics should be shown. Finally, an efficient parallel implementation must be developed in order to compete with state-of-the art operational models.

    A key element to the robustness and accuracy of any AMR method is its refinement criterion, which decides when and where

to coarsen or refine the computational grid. Ferguson et al. (2016) evaluate several *ad hoc* criteria and identify appropriate ones for their numerical experiments. However they do not find any "clear strategy for establishing the best general refinement criteria." In contrast, `wavetrisk` uses objective and clearly defined refinement criteria which control the multiscale relative error of the solution or of its tendencies as measured directly by the wavelet coefficients.

    In the following section we review briefly the basic properties and computational features of the wavelet-based dynamical

adaptivity laid out in Dubos and Kevlahan (2013) and Aechtner et al. (2015). Section 3 summarizes the discrete equations



solved by the model and presents the ALE vertical coordinate. Section 4 applies the principle of wavelet-based adaptivity to present the context. In section 5, implementation details are given and the adaptive solver is evaluated using three test cases of increasing complexity. Section 6 concludes.

## 2 The adaptive wavelet method

### 2.1 Wavelet adaptivity on the plane

The foundations for `wavetrisk` were set out in Dubos and Kevlahan (2013) for the shallow water equations on the $\beta$ plane. We used the TRiSK discretization scheme on the hexagonal–triangular C-grid proposed by Ringler et al. (2010) because of its excellent mimetic properties. The principal goal of our approach was that the adaptivity should be an overlay on the flux-based discretization. Mimetic properties (e.g. mass conservation) should be preserved by the adaptivity and the discretizations should be unchanged. The building blocks of the method are one-scale operators (in this case, the TRiSK discretization) and two-scale operators between a fine scale $j+1$ and a coarse scale $j$. To conserve the mimetic properties of the TRiSK scheme the restriction operators from scale $j+1$ to scale $j$ and the discrete differential operators (div, grad, curl) must satisfy the following commutation properties

$$R_\mu^j \circ \mathrm{div}^{j+1} \quad = \mathrm{div}^j \circ R_F^j \qquad \textit{conserves mass}, \tag{1}$$

$$\mathrm{curl}^j \circ R_\mathbf{u}^j \quad = R_\zeta^j \circ \mathrm{curl}^{j+1} \quad \textit{conserves circulation}, \tag{2}$$

$$\mathrm{grad}^j \circ R_B^j \quad = R_\mathbf{u}^j \circ \mathrm{grad}^{j+1} \quad \textit{no spurious vorticity}, \tag{3}$$

where $R_\mu$ is mass density (or height) restriction, $R_F$ is the flux restriction, $R_u$ is the velocity restriction, $R_\zeta$ is the circulation (vorticity) restriction, and $R_B$ is the Bernoulli function restriction. The third commutation relation ensures that a flow with uniform potential vorticity remains uniform under the advection by an arbitrary velocity field (i.e. vorticity is advected like a tracer).

The C-grid is a staggered grid where vorticity is located on triangles (the primal grid), and mass is located on the hexagons formed from the bisectors of the triangle edges (the dual grid). Velocity is located on the edges of triangles, which are also the perpendicular bisectors of the hexagonal edges. (Note that we have chosen the opposite notation to Ringler et al. (2010) and Dubos et al. (2015) since in the multiscale case the triangle grid is generated first by repeated bisection from the icosahedron and the hexagon grid is generated as the dual grid of the triangles.) The regular C-grid is shown in figure 1. Starting with a coarsest grid, a multiscale hierarchy of primal grids is constructed by bisection of the triangle edges. The dual grid of hexagons is constructed from the perpendicular edge bisectors of the primal grid. On the plane, both the hexagons and triangles are nested and regular. We will see below that this is not the case for the sphere.

This hierarchy of computational grids leads to a so-called wavelet multiresolution analysis (MRA), i.e. the nested sequence of approximation subspaces used to construct the second-generation biorthogonal wavelets (Sweldens, 1998) that are the basis of the grid adaptivity algorithm. The MRA provides a sequence of smooth approximations of a function $f(x)$ on each grid level $j$ and an associated sequence of "details" which give the differences between the approximation at a fine level $j+1$ and





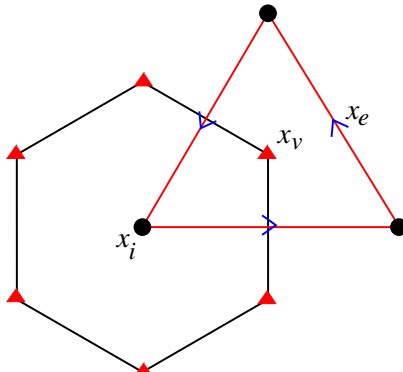

**Figure 1.** The regular C-grid on the plane. Vorticity is located on the primal grid of triangles at points $x_v$ and mass located on the dual grid of hexagons at nodes $x_i$. The velocities/fluxes are located on edges $x_e$. A multiscale hierarchy of nested refined C-grids is generated by bisecting the triangle edges.

a coarse level $j$. The details are effectively the interpolation errors between scales $j+1$ and $j$. The smooth approximation at scale $j$ has a basis of *scaling functions* $\{\phi_k^j(x)\}$, while the details between scales $j+1$ and $j$ has a basis of *wavelets* $\{\psi_k^j(x)\}$.

If the wavelet coefficient at a particular position $k$ and scale $j$ is sufficiently small, i.e. $|\psi_k^j(x)| < \varepsilon$, then the value of $f(x)$ is well-approximated by the function $f_\geq(x)$ interpolated from neighbouring scaling function values at the same scale. If the

5   wavelet coefficient is large, the value of the wavelet coefficient (i.e. the details) must be retained and added to the interpolated value to obtain an sufficiently accurate approximation. Neglecting the small wavelet coefficients (and associated grid points) generates a multiscale hierarchy of adapted grids. The commutation relations (2–3) ensure that the mimetic properties of the discretization are also satisfied on the adapted grid.

One can prove that this nonlinear wavelet filtering provides error control,

$$||f(x) - f_\geq(x)||_\infty \quad = \quad O(\varepsilon), \tag{4}$$
$$\mathcal{N} \quad = \quad O(\varepsilon^{-1/2N}), \tag{5}$$
$$||f(x) - f_\geq(x)||_\infty \quad = \quad O(\mathcal{N}^{-2N}), \tag{6}$$

where $\mathcal{N}$ is the number of grid points retained on the adapted grid and $N$ is the order of interpolation. Thus, we can also estimate how many grid points (i.e. computational elements) are required to obtain an approximation with a desired error $\varepsilon$.

15   Setting $\varepsilon$ therefore determines the numerical error of the approximation (the tolerance) and also determines the number of grid points in the adapted grid. In principle, it is not necessary to set a maximum resolution scale $J$ since it is determined automatically by $\varepsilon$. Note that an approximation $f_\geq^j(x)$ at each scale $j$ is provided by the set of scaling functions $\{\phi_k^j\}_\geq$ and their coefficients.

In previous work we have shown that the cpu time $\tau_n$ per time step is directly proportional to the number of active grid points

20   $\mathcal{N}$. Combined with the error estimates (6), this shows that adaptive wavelet method is efficient in the sense mentioned in the introduction: the cpu time scales with the specified error and the error is controlled regardless of the structure of the flow. This





is not true for non-adaptive methods where there is no explicit link between the actual numerical error and the number of grid points. At best, the approximation error of the discretization on a particular static grid gives only an upper bound on the actual error. Even this upper bound may not be satisfied if the *a priori* estimate of the smallest structures (e.g. strongest gradients) that determined the chosen grid resolution did not properly take into account intermittency and rare events. For example, Yakhot

and Sreenivasan (2005) argue that temporal and spatial intermittency mean that turbulent flows require far higher resolution than that found using the usual Kolmogorov scale based estimate, $\Delta x \sim \mathrm{Re}^{-3/4}$.

The above adaptive algorithm controls the error of the solution at each time step, but it does not allow for dynamics. The solution can change over one time step from $t^n$ to $t^n + \Delta t$ by translating at the same scale, coarsening (gradient weakens) or refining (gradient strengthens). If the CFL criterion is one, i.e. $\Delta t < \Delta x_{\min}/||u||_\infty$, the solution can translate by a maximum

of one grid point at the same scale over one time step. If the nonlinearities in the governing equations are quadratic, the active scale can increase by a maximum of a factor of two from $j$ to $j+1$ over one time step. To allow for these changes, an "adjacent zone" is added to the set of active wavelet coefficients $\{\psi_k^j\}_\geq$ that includes its nearest neighbours in both position and scale (Liandrat and Tchamitchian, 1990). The adaptive grid must also satisfy the perfect reconstruction criterion: there must be sufficient scaling functions (grid points) to construct the wavelets. There must also be sufficient grid points present at

each scale $j$ to construct the required TRiSK differential operators (by interpolation, if their associated wavelets are inactive). Once the new adapted grid has been constructed the prognostic variables are interpolated onto the new grid by performing an inverse wavelet transform. After the solution is advanced, the wavelets on the union of the adapted grid and adjacent zone are again filtered using the threshold $\varepsilon$ to obtain a new set of active wavelets at time $t^n + \Delta t$.

Because we use a staggered grid, the adaptive wavelet algorithm described above differs fundamentally from previous

adaptive wavelet collocation methods (e.g. Mehra and Kevlahan, 2008; Kevlahan and Vasilyev, 2005; Roussel and Schneider, 2010; Schneider and Vasilyev, 2010). Since mass and velocity are located at different points, we must construct two distinct wavelet transforms: a scalar-valued wavelet transform for mass density $\mu$ and a vector-valued wavelet transform for velocity $u$. To control the errors in the tendencies the corresponding thresholds $\varepsilon_\mu$ and $\varepsilon_u$ must be properly scaled. The details of these scalings in the inertia–gravity wave and geostrophic regimes are given in Dubos and Kevlahan (2013).

Finally, note that ideally the time step should also adapt to the local grid scale. In other words, the solution at each scale should be advanced on the time step $\Delta t^j$ appropriate for that scale. Although this scale-dependent time stepping is optimal, in practice it does not provide much advantage unless only a small portion of the total active grid points is at the finest scale, which is not usually the case. Domingues et al. (2008) developed a second-order Runge–Kutta scale-dependent time stepping and McCorquodale et al. (2015) extended scale-dependent time stepping to arbitrary order. For simplicity, we have decided not

to implement scale-dependent time stepping in `wavetrisk`, although it could be added in the future.

The ability of the adaptive wavelet method to control the errors of the tendencies was verified in Dubos and Kevlahan (2013) and the computational performance and parallel efficiency of the method on the sphere were confirmed in Aechtner et al. (2015). Since the three-dimensional hydrostatic code uses the multilayer shallow equations and horizontal adaptivity only, these properties are inherited by `wavetrisk`.





| $J$ | $N$ | $\overline{\Delta x}$ (deg) | $\overline{\Delta x}$ (km) |
|---|---|---|---|
| 0 | 12 | 65.9 | 7 317 |
| 1 | 42 | 34.1 | 3 790 |
| 2 | 162 | 17.2 | 1 913 |
| 3 | 642 | 8.6 | 960 |
| 4 | 2 562 | 4.3 | 480 |
| 5 | 10 242 | 2.2 | 240 |
| 6 | 40 962 | 1.1 | 120 |
| 7 | 163 842 | 0.54 | 60 |
| 8 | 655 362 | 0.27 | 30 |
| 9 | 2 621 442 | 0.14 | 15 |
| 10 | 10 485 762 | 0.068 | 7.5 |

**Table 1.** Hierarchy of multiscale primal (triangle) grids derived by edge bisection from the icosahedron at scale $J = 0$. $N = 10 \times 4^J + 2$ is the number of computational elements (lozenges), $\overline{\Delta x} = (4/\sqrt{3}4\pi^2 a^2/20/4^J)^{1/2}$ is the average triangle edge length on the Earth. Each computational element is made up of one node (for scalars), three edges (for velocities) and two triangles (circulation). Therefore the total number of data elements is $5N$ per vertical level. Note that there are two exceptional points to deal with the poles when the icosahedron is unfolded into ten lozenges. Typically the coarsest (optimized) level is $4 \le J \le 9$ with three to five levels of refinement.

The prototype `matlab` serial implementation of the adaptive algorithm in the plane is relatively straightforward because the computational grid is uniform and we did not parallelize the algorithm. In the following section we review the extension of the algorithm to the sphere.

### 2.2 Wavelet adaptivity on the sphere

We have reviewed the essential elements of the adaptive wavelet method in the previous section, however to be practically useful for a dynamical core the method must be extended to the sphere and parallelized. The extension to the sphere presents numerous technical challenges. First, the grid is non-uniform, each computational element is geometrically unique and the icosahedral grid on the sphere has 12 pentagonal dual grid cells (a sphere cannot be tiled uniformly). Secondly, the repeated bisection of the primal triangular grid used to generate the multiscale grid structure produces a grid that is increasingly distorted

near the edges of the original icosahedron. Finally, the dual hexagonal grid is no longer nested between successive scales because the triangles used to generate it are not equilateral. This makes it complicated to construct the flux restriction from fine to coarse scales because we must keep track of small overlapping regions. In the following, we focus on the most important modifications necessary to deal with the non-uniform geometry and to parallelize the code in the following.

The wavelet transform on the non-adaptive primal (triangular) icosahedral grid is shown in figure 2. Table 1 gives the

properties of the multiscale grids for each level of refinement $J$.



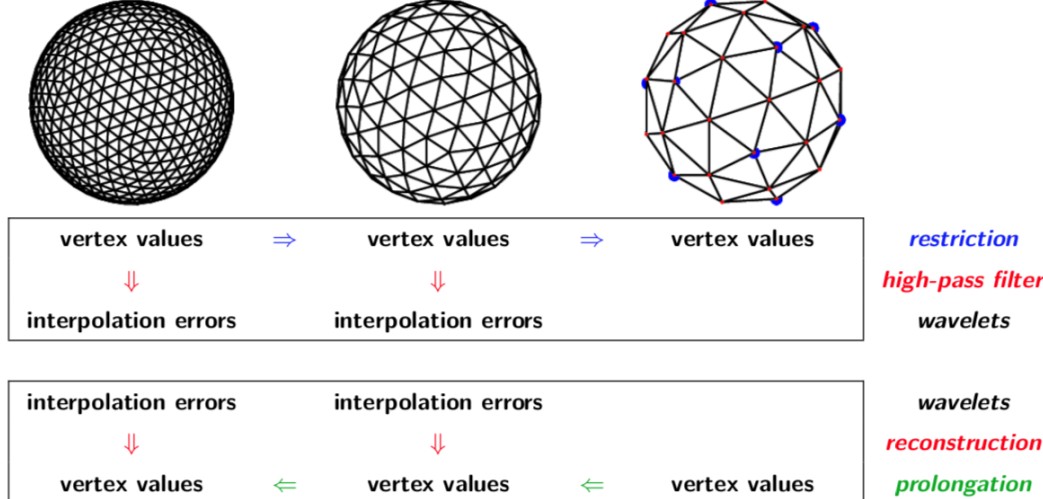

**Figure 2.** Wavelet transform on a non-adaptive icosahedral grid with three scales.

The basic principles of the adaptive transforms are the same as on the plane, however we must take account of the fact that the geometry of the grid is irregular. The TRiSK discretization Ringler et al. (2010) is second-order accurate for equilateral triangles, but drops to first-order accurate when the triangles are far from equilateral. This problem is not restricted to adaptive methods, but is an issue for all methods on fine icosahedral grids. To deal with this problem we first generate a coarsest grid

(e.g. $J_{\min} = 5$ levels of bisection) and then optimize its geometry to ensure that the bisection of primal and dual edges is as close as possible. This optimized primal grid is then bisected as needed to generate the multiscale grids required for the wavelet transform. The default method is to read in optimized grids provided by Heikes et al. (2013). As an alternative, we can also use the grid optimization proposed by Xu (2006), although it produces less optimal grids.

A more fundamental challenge particular to this adaptive flux-based method on staggered grids is that the dual hexagonal

grids on two successive scales are no longer nested as they are on the plane. This means that we must keep track of the various configurations of small overlapping hexagons when computing the flux restriction (see section 4.3 and figure 4 of Aechtner et al. (2015) for more details). Note that the primal grid of triangles remains nested on the sphere, which means that the restrictions of velocity, Bernoulli and circulation and straightforward.

## 2.3 Parallelization and data structure

The parallelization and data structure must take account both the icosahedral geometry and topology of the spherical discretization and the fact that the grid is adaptive and multiscale. The C-grid is stored as a regular data structure by grouping one node (mass and other scalars), two triangles (circulations) and three edges (velocities) into one computational element, a "lozenge" as shown in figure 3. The icosahedron is composed of 20 triangles grouped into 10 lozenges. Therefore, a grid



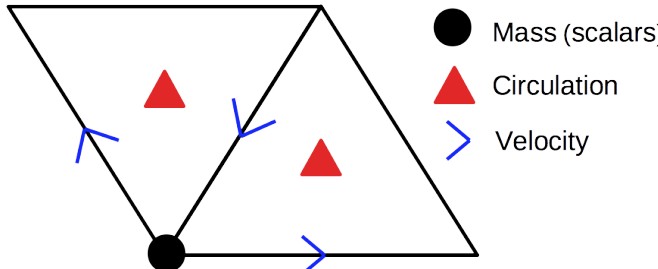

**Figure 3.** Lozenge: basic computational element containing one node (for mass and other scalars), three edges (for velocities) and two triangles (for circulation). Separate wavelet transforms are provided for the nodes (scalar-valued) and edges (vector-valued). The adaptive grid consists of the the significant nodes and edges, together with nearest neighbours in position and scale necessary for dynamics.

resulting from refining an icosahedron can be divided into 10 sub-grids each of which can be stored and accessed in a regular fashion. Note that at the edges of the lozenges the two adjacent regular grids of the original icosahedron are rotated with respect to each other. This is dealt with by surrounding the 10 lozenge sub-domains by ghost/halo cells. The halo cells are also used for parallel communication during boundary updates between cores.

5     The most natural way to store two-dimensional data with a dyadic multiscale structure is to use a quad tree, where each branching represents a new finer scale. However, in an adaptive method some branches are pruned and accessing neighbours would require wasteful traversing of the tree structure. We decided to use a hybrid data structure where each quad tree terminates in a patch. The patches are small $4 \times 4$ or $8 \times 8$ regular grids. This structure reduces the number of levels in the quad tree and makes it more computationally efficient to find neighbours. A similar hybrid approach was used by Behrens (2009) 10  and Hejazialhosseini et al. (2010). Using patches increases memory slightly because inactive elements are stored. It is also possible to try to optimize the patch size for best computational performance for a particular problem, although $8 \times 8$ appears optimal in most cases. Note that the patches do not affect the results of the computation, they just improve computational efficiency.

    The domain decomposition used for parallelizing the computation is based on distributing the lozenge sub-domains on the 15  coarsest level $J_{\mathrm{min}}$ (and their associated children) to different cores. Each core can compute several sub-domains and having several small sub-domains per core can improve cache efficiency. Note that there are $10 \times 4^{J_{\mathrm{min}} - (P+1)}$ sub-domains at the coarsest level with a patch size of $2^P \times 2^P$. For example, if $J_{\mathrm{min}} = 7$ and $P = 2$ the coarsest scale contains $N_D = 2560$ sub-domains. Because the number of cores $N_{\mathrm{core}} \leq N_D$ large numbers of cores are usable only by large $J_{\mathrm{min}}$.

    In an adaptive simulation each sub-domain typically has a different number of active computational elements, and thus gen- 20  erates different loads for communication and computation. This load imbalance between cores can seriously degrade parallel performance. To remedy this we use a simple rebalancing algorithm to redistribute sub-domains amongst the cores to produce a more balanced load. This rebalancing is done at each checkpoint save.

    Every sub-domain is extended to hold as many ghost/halo cells as necessary for the various required operators. The values at the halo cells are communicated as needed. Intra-core communication is done by copying and inter-core communication is





done using `mpi`. During grid adaption new patches are added and removed as required and grid connectivity between domains is updated (via `mpi` as necessary). Critical communications are carried out locally point-to-point rather than using global communication where possible. Where possible communication is non-blocking so that the computations can continue while communication is taking place in the background.

This parallelization algorithm is reasonably efficient for at least several hundred or a few thousand cores. Dubos and Kevlahan (2013) found that good weak parallel efficiency is possible with as few as 1300 computational elements per core in adaptive runs. The three-dimensional code has better parallel efficiency because the column structure of the data produces a higher computational load for each active grid element. We present some representative strong and weak parallel scaling results in section 5.1.

## 3   Hydrostatic dynamical equations and ALE vertical coordinates

As mentioned in the introduction, we use the `dynamico` discretization of the three-dimensional hydrostatic multilayer shallow water equations Dubos et al. (2015) in compressible form. The discrete dynamical equations are derived from the discrete Hamiltonian, which allows the construction of energy or potential enstrophy conserving equations. The prognostic variables are $m_{ik}$ (mass), $\Theta_{ik}$ (mass-weighted potential temperature) and $v_{ek}$ (velocity), where $k$ labels a full vertical level, $l$ an in-
terface (half-level) between full vertical levels, $i$ an hexagonal or pentagonal cell, $v$ a triangle and $e$ an edge. In terms of the Hamiltonian, their evolution equations are :

$$\frac{\partial m_{ik}}{\partial t} + \delta_i \frac{\partial H}{\partial v_{ek}} = 0, \qquad\qquad \frac{\partial \Theta_{ik}}{\partial t} + \delta_i \left( \theta_{ek}^* \frac{\partial H}{\partial v_{ek}} \right) = 0, \tag{7}$$

$$\frac{\partial v_{ek}}{\partial t} + \left( \frac{f_v + \delta_v v_k}{\overline{m_{ik}}^v} \frac{\partial H}{\partial v_{ek}} \right)^{\perp} + \delta_e \frac{\partial H}{\partial m_{ik}} + \theta_{ek}^* \delta_e \frac{\partial H}{\partial \Theta_{ik}} = 0, \tag{8}$$

$$\qquad\quad \text{potential vorticity} \qquad\quad \text{Bernoulli} \qquad \text{Exner} \tag{9}$$

where $\delta_i$, $\delta_e$ and $\delta_v$ are discrete divergence, gradient and curl operators yielding values at cells, edges and triangles respectively, $f_v$ is the Coriolis parameter, and $\theta_{ek}^*$, $\overline{m_{ik}}^v$ are values of $\theta$ and $m$ reconstructed at edges and triangles, respectively, by appropriate averaging. Indices than can be inferred may be omitted, as in $\delta_v v_k \equiv \delta_v v_{ek}$. Evaluating the Hamiltonian terms and discretizing, we obtain the inviscid discrete dynamical equations

$$\frac{\partial \mu_{ik}}{\partial t} + \delta_i U_k \;\; = \;\; 0, \tag{10}$$

$$\frac{\partial \Theta_{ik}}{\partial t} + \delta_i (\theta_{ek}^* U_k) \;\; = \;\; 0, \tag{11}$$

$$\frac{\partial v_{ek}}{\partial t} + \delta_e B_k + \theta_{ek}^* \delta_e \pi_k + (q_k U_k)_e^{\perp} \;\; = \;\; 0, \tag{12}$$

where $\mu_{ik} = \rho_{ik} \Delta z_{ik}$ and we have assumed Lagrangian vertical coordinates (so the vertical mass fluxes are not present). Potential temperature $\theta_{ik} = \Theta_{ik}/\mu_{ik}$ and $\theta_{ek} = \overline{\theta_{ik}}^e$. $\mu_{ik}$ is a pseudo mass density (equivalent to $\rho_{ik} \Delta z_{ik}$), $U_{ek}$ is the horizontal mass flux, $B_{ik}$ is the Bernoulli function, $\pi_{ik} = \pi(\alpha_{ik}, \theta_{ik})$ is the Exner function, $\alpha_{ik}$ is the specific volume, and $q_{vk}$ is the





potential vorticity. In the compressible case we consider here, the Bernoulli function is given by

$$B_{ik} = K_{ik} + \overline{\Phi_{il}}^k, \tag{13}$$

where $K_{ik}$ is the discrete kinetic energy computed from $u_{ek}^2$ using appropriate averaging, and $\Phi_{il}$ is the geopotential at vertical layer interfaces $l$. The discrete operators $\delta_i$ (divergence with result at a node), $\delta_e$ (gradient with result at an edge) and $(\cdot)^\perp$

(perpendicular flux) are defined as in Ringler et al. (2010) and the shallow water equations version of `wavetrisk`.

Each evaluation of the trend requires integrating vertically down to find the surface pressure and then integrating up to find the Exner function and geopotential (and hence the Bernoulli function).

The choice of Lagrangian vertical coordinates (rather than mass-based) is simple, computationally efficient and especially well suited to ocean modelling since it virtually eliminates numerical vertical diapycnal diffusion, unlike a $z-$coordinate.

This will become important when we develop the ocean version of `wavetrisk` (see Kevlahan et al., 2015). The vertical coordinates are pressure-based and may either be evenly spaced or hybrid ($\sigma$). As the flow develops the vertical levels expand and contract which can lead to loss of accuracy, or even negative mass (if a layer collapses to zero thickness). To avoid this problem, we remap the vertical coordinates back to the original coordinates either every time step or or every ten time steps or so. The remapping takes about 7 % of cpu time if done every time step.

Remapping Lagrangian vertical coordinates is a common technique in ocean modelling (e.g. Petersen et al., 2015), where the target grid is updated at each time step to optimize numerical accuracy and stability (e.g. the target grid may be based on approximately isopycnal or isentropic coordinates) and is referred to as an "arbitrary Lagrangian–Eulerian" (ALE) coordinate system. Optimizing the target grid in `wavetrisk` would add a sort of vertical $r-$adaptivity in addition to the wavelet-based horizontal $h-$adaptivity, where the number of vertical levels would remain constant but their locations would be chosen op-

timally. For example, Kavcic and Thuburn (2018) choose the target grid levels at each time step to keep the vertical levels close to isentropic. This requires solving a small elliptic optimization problem at each grid level. In principle, it is also possible to include vertical $h-$adaptivity by locally de-activating certain vertical layers if the vertical interpolation errors are small. We have tested `wavetrisk` with a variety of piecewise constant, piecewise linear, piecewise parabolic and piecewise quartic remapping schemes, modified from packages supplied by Shchepetkin (2001) and Engwirda and Kelley (2016,

https://github.com/dengwirda/PPR). The user can select no limiter, a monotone limiter or a sWENO limiter. Any of these remapping algorithm can be selected simply by changing a parameter.

Mass density $\mu$, potential temperature $\theta$ and velocities $u, v, w$ are remapped onto the original hybrid pressure coordinates. This remapping scheme conserves mass, potential temperature and divergence and appears to perform well. We also tested Lin (2004)'s scheme, which remaps momentum and total energy, but it proved to be less stable.

We found that simple piecewise constant remapping gives qualitatively incorrect results for zonally averaged statistics in the Held and Suarez (1994) test case, but that it is sufficient for the mountain-induced Rossby wave and baroclinic instability test cases. Piecewise linear and piecewise parabolic remapping give qualitatively accurate results. The target grid could be optimized at each remap, as in ocean models and as explored by Kavcic and Thuburn (2018) for the `endgame` GCM. However, the current procedure gives good results in the test cases we have examined.



The dynamical equations are advanced in time using the fourth-order four-stage low storage Runge–Kutta routine used in `dynamico` (Dubos et al., 2015). Various strong stability preserving schemes (e.g. the third-order three-stage scheme RK33ssp, RK45ssp) are also available as options (Spiteri and Ruuth, 2002).

In many cases the code runs stably without any additional diffusion or filtering of small scales. However, in rare cases the code crashes due to numerical instability. In order to improve stability a regular or second order hyper-diffusion term is added to the dynamical equations for the prognostic variables to damp the largest wavenumbers (both the divergent and vortical modes of the momentum equation are damped). Using the discrete operator notation of Dubos et al. (2015) the diffusion terms for scalars, divergence and vorticity are, respectively

$$D_\phi = K_\phi \delta_i \left[ \frac{l_e}{d_e} \delta_e \left( \frac{\phi}{A_i} \right) \right], \tag{14}$$

$$D_\delta = K_\delta \delta_e \left[ \frac{1}{A_i} \delta_i \left( \frac{l_e}{l_d} v_e \right) \right], \tag{15}$$

$$D_\omega = K_\omega \delta_e \left[ \frac{1}{A_v} \delta_v (v_e) \right], \tag{16}$$

where $d_e$ is a triangle edge length (primal grid), $l_e$ is a hexagon edge length (dual grid), $A_i$ is a hexagon area, $A_v$ is a triangle area. These discretizations correspond to the continuous differential operators $\nabla \cdot \nabla (\phi)$, $\nabla (\nabla \cdot \mathbf{u})$ and $\nabla \times (\nabla \times \mathbf{u})$ respectively.

A general $p$-th order hyperdiffusion operator is defined as an iterated Laplacian operator $\Delta^p$ corresponding to $D_\phi^p$, $D_\delta^p$, $D_\omega^p$. We choose either $p = 1$ (regular diffusion) or $p = 2$ hyperdiffusion. Diffusion may be applied at each time step, or every $N_{\text{diff}} > 1$ time steps (where $N_{\text{diff}}$ is limited by viscous stability in time). The diffusion coefficient $K$ is chosen to give the same amount of damping over a time step,

$$K = \frac{\overline{\Delta x}^{2p}}{\Delta t} C N_{\text{diff}}, \tag{17}$$

where $\overline{\Delta x} = (4\pi a^2 / (10 \, 4^J + 2))^{1/2}$ is the average grid scale and $C$ is an empirical constant chosen to ensure stability, which may be different for different variables.

Note that in addition to ensuring stability, adding diffusion can improve the efficiency of the adaptivity by damping out small fluctuations that might otherwise produce some local grid refinement. This effect is especially significant when adapting on the wavelets of the tendencies. We present results both with and without explicit diffusion.

## 4 Adaptivity

To extend the adaptive algorithm for the two dimensional shallow water equations reviewed in section 2.1 to the three-dimensional case we simply apply the two-dimensional algorithm to each vertical layer in turn and then define the adapted grid to be the union of the adapted grids over all vertical layers $k = 1, \ldots, N$.

After the time step and vertical re-gridding have been completed, the wavelet transforms of mass density $\mu$, mass-weighted potential temperature $\Theta$ and velocity $u$ are calculated for each vertical level. Then, grid points are labeled as active if their asso-





ciated wavelet coefficient has a magnitude greater than or equal to the appropriate relative error threshold. Nearest neighbours are added to the adjacent zone in position and scale. Grid points are added as necessary to satisfy the perfect reconstruction criterion (so the grid points necessary to compute the wavelet coefficients are present). Finally, grid points required for the TRiSK operators are labelled. An inverse wavelet transform interpolates the solution conservatively onto the new adapted grid.

The adaptivity algorithm is summarized in algorithm 1.

The adaptive algorithm produces an adapted grid consisting of vertical columns of varying horizontal size. In practise, a maximum scale $J$ is usually set based on available computational resources and user requirements. This "column adaptivity" approach is not optimal for vertically tilted structures, but it provides much better load balancing and is far simpler than dealing with fully three-dimensional adaptivity. In addition, we can take advantage of an ALE formulation for the vertical

coordinate which is often more flexible and accurate than a $z$-coordinate system. We show in the results section that column adaptivity provides accurate results and good grid compression ratios. Note that the vertical grid is remapped (if necessary) before adapting the horizontal grid.

The normalizations for the absolute tolerances $\varepsilon_\mu = \varepsilon||\mu||_\infty$, $\varepsilon_\Theta = \varepsilon||\Theta||_\infty$ and $\varepsilon_u = \varepsilon||u||_\infty$ determining the grid adaptation may be defined either *a priori* by dimensional analysis and knowledge of the flow being simulated, or they are determined

dynamically by estimating the appropriate norms separately for each vertical layer.

The choice of normalization for the absolute thresholds $\varepsilon_\mu$, $\varepsilon_\Theta$ and $\varepsilon_u$ is a crucial and sensitive part of the algorithm since they guarantee the relative accuracy and efficiency of the method. In the shallow water case we determined the tolerances for mass and velocity based on dimensional analysis of the tendencies in the inertia-gravity and quasi-geostrophic regimes. This sort of dimensional analysis is not feasible for the three-dimensional equations so we have developed two strategies to ensure

uniform control of the relative error in the tendencies of the prognostic variables in each vertical layer.

As we mentioned earlier, in the first approach these tolerances are set by dimensional analysis using a knowledge of the appropriate scales for the test problem under consider. This gives reasonable results for problems with fairly stationary evolution, but can lead to less accurate results if the dynamics change significantly during the course of the simulation. It also generally means that the tolerances set to the same value for all vertical levels.

In the second approach, the relevant norms are calculated dynamically at each time step and at each vertical level to ensure they are consistent with the actual state of the flow. This approach is more generally applicable and does not require the user to have any *a priori* knowledge of the solution.

In addition to determining how to calculate the normalization for the thresholds, we also need to decide which wavelets we are filtering. In the shallow water case we always directly filtered the wavelets of the solution, which measure the interpolation

errors of the variables themselves between two grid levels at a given time step. Although this does not directly control the tendency error, analysis of a linearized problem suggests that it should provide some control of the tendency error provided the solution is smooth.

Consider a set of $n$ coupled linear ordinary differential equations

$$\frac{du}{dt} = A(u), u(0) = u_0, \tag{18}$$





---

**Algorithm 1** Adaptive grid algorithm. Executed after the time step and vertical grid remapping is complete. Note that wavelets are calculated and filtered at all vertical levels so final adapted grid is union of adapted grids over all vertical levels.

---

After time step and vertical grid remapping have been completed

**if** adapt on trend **then**

    Compute trend

    Compute trend wavelets

**end if**

**for** k = 1 **to** N **do** {loop over all vertical levels}

    Update relative error thresholds $\varepsilon_\mu, \varepsilon_\Theta, \varepsilon_u$ by computing appropriate norms at each vertical level

**end for**

Label all active nodes and edges at coarsest scale $j = J_{\min}$ as **adjacent zone**

Label all nodes and edges for scales $j > J_{\min+1}$ as **inactive**

**for** $k = 1$ **to** $N$ **do** {check all vertical levels}

    **for** $j = J - 1$ **to** $J_{\min}$ **do** {check wavelets at all scales}

        label as **active** all nodes with an associated wavelet coefficients $\geq \varepsilon_\mu$ OR $\geq \varepsilon_\Theta$

        label as **active** all edges with an associated wavelet coefficients $\geq \varepsilon_u$

    **end for**

**end for**

Add **nearest neighbours** of active grid points at **coarser scale** to **adjacent zone**

Add **nearest neighbours** of active grid points at **same scale** to **adjacent zone**

Add/remove **patches** as required for neighbours at finer scales

**for** $j = J - 1$ **to** $J_{\min}$ **do**

    add neighbours of active grid points at **finer scales**

**end for**

**Perfect reconstruction criterion**: add grid points required to compute active and adjacent zone wavelets

Label grid points required by **TRiSK operators** to compute trend at active and adjacent zone wavelets

Set all **wavelets** not active or in the adjacent zone to zero

**Inverse wavelet transform** of solution onto new adapted grid

---




where $A$ is an $(n \times n)$ constant coefficient matrix. Applying an Euler scheme gives the time step $u^{n+1} = u^n + \Delta t \, A u^n$. The error (wavelet coefficient) $\tilde{u}^n$ satisfies the same equation, so we have

$$
\begin{aligned}
\tilde{u}^{n+1} &= (I + \Delta t A)\tilde{u}^n, \\
\frac{||\tilde{u}^{n+1}||}{||u^n||} &\leq (1 + \Delta t ||A||)\frac{||\tilde{u}^n||}{||u^n||},
\end{aligned}
$$

where $I$ is the identity matrix and we have used the triangle inequality. Now, if we assume that $||\tilde{u}^n||/||u^n|| \leq \varepsilon$ by wavelet filtering of $u^n$, we have the following bound on the relative error

$$
\frac{||\tilde{u}^{n+1}||}{||u^n||} \leq \varepsilon(1 + \Delta t ||A||). \tag{19}
$$

If $A$ is symmetric then $||A||_2 = \rho(A)$ where $\rho(A)$ is the spectral radius of $A$ (largest magnitude eigenvalue of $A$, $|\lambda|_{\max}$). Thus, we have that

$$
\frac{||\tilde{u}^{n+1}||_2}{||u^n||_2} \leq \varepsilon(1 + \Delta t \, \rho(A)). \tag{20}
$$

However, a necessary condition for stability of the Euler method is that

$$
\Delta t \, \rho(A) \leq 2. \tag{21}
$$

Which gives

$$
\frac{||\tilde{u}^{n+1}||_2}{||u^n||_2} \leq 3\varepsilon. \tag{22}
$$

We have therefore shown that wavelet filtering of the solution itself provides control of the relative error of the solution over one time step for symmetric discretizations. In general, for non-symmetric $A$ we have

$$
\frac{||\tilde{u}^{n+1}||_2}{||u^n||_2} \leq \varepsilon(1 + \Delta t \, \sigma(A)). \tag{23}
$$

where $\sigma(A)$ is the largest singular value of $A$. However, there exists an $\epsilon > 0$ such that $||A|| \leq \rho(A) + \epsilon$ for any matrix norm $||\cdot||$. Thus, we can expect similar relation to (22) to hold for more non-symmetric discretizations. Although we have only

considered the linear constant coefficient case, the results suggest that dynamic wavelet filtering of the solution, together with dynamic calculation of the normalization of the thresholds, should control the relative error of the solution over one time step.

Now, consider filtering on the wavelets of the tendencies $T$ themselves at the previous time step so that $||\tilde{T}^n||/||T^n|| \leq \varepsilon$ where $\tilde{T}$ is now the error (wavelet coefficient) of the tendency. To first order in $\Delta t$ we have

$$
\tilde{T}^{n+1} = \tilde{T}^n + \Delta t \left( J_T(\tilde{u}^n, t^n)\tilde{T}^n + \frac{\partial T}{\partial t}(\tilde{u}^n, t^n) \right), \tag{24}
$$

where $J_T(\tilde{u}^n, t^n)$ is the Jacobian matrix of the tendency evaluated at the previous time step. In the case of an Euler method applied to a linear constant coefficient system of ordinary differential equations as we considered above, we have $T = Au$ and



the errors in the tendency $\tilde{T}$ (wavelets of the tendency) satisfy

$$
\begin{aligned}
\tilde{T}^{n+1} &= (I + A)\tilde{T}^n, \\
||\tilde{T}^{n+1}|| &\leq ||\tilde{T}^n|| + ||A\tilde{T}^n||, \\
||\tilde{T}^{n+1}|| &\leq ||\tilde{T}^n|| + ||A||\,||\tilde{T}^n||,
\end{aligned}
$$

where we have used the triangle and Schwartz inequalities. Normalizing by $||T^n||$ and recalling that wavelet filtering ensures that $||\tilde{T}^n||/||T^n|| \leq \varepsilon$ gives the result

$$
\frac{||\tilde{T}^{n+1}||}{||T^n||} \leq (1 + ||A||)\,\varepsilon. \tag{25}
$$

Again, if $A$ is symmetric, we have $||A||_2 = \rho(A)$ and using $\rho(A) < 1/\Delta t$ for the Euler method we find

$$
\frac{||\tilde{T}^{n+1}||}{||T^n||} \leq (1 + \frac{1}{\Delta t})\varepsilon \leq C\varepsilon. \tag{26}
$$

Inequality (26) shows that filtering the wavelets of the tendencies $\tilde{T}^n$ from the previous time step effectively controls the relative tendency error in the next time step up to a constant factor depending on the discretized system. Note that filtering the wavelets of the tendencies is slightly more expensive since it requires an additional evaluation of the tendencies (or an additional inverse wavelet transform).

In the following section we apply `wavetrisk` to solve three standard test problems for hydrostatic dynamical cores. The emphasis is on evaluating the adaptivity in the three-dimensional hydrostatic version, rather the basic accuracy of the method since the underlying discretization is the same as `dynamico` and we have already assessed the basic features of the error control in previous work on the shallow water equations. We directly compare the options of nonlinear filtering the wavelets of the solution and wavelet filtering the wavelets of the tendencies. Filtering the tendencies is more precise but more sensitive to
the choice of threshold.

## 5    Validation test case results

### 5.1    Parallel and computational efficiency

The adaptive `wavetrisk` code has significant computational overhead compared to `dynamico`, which solves the same discretized equations. This overhead is required to deal with the local geometry of the grid (which is not pre-computed), the
multiscale grid structure and the parallel communication on the hybrid data structure. This overhead increases with the number of refinement levels and decreases for larger patch sizes. A lower bound on the overhead can be estimated by directly comparing wall clock time for `wavetrisk` and `dynamico` since both codes are based on the same TRiSK discretization. The codes solved DCMIP 2008 test case 8 (see section 5.2) on the uniform grid $J = 6$ (1° degree) with 27 Lagrangian vertical levels and $8 \times 8$ patches for `wavetrisk`. The codes were run on 160 cores. This limited test suggests that `wavetrisk` is approximately



50% slower than `dynamico` per active grid point. The actual overhead depends on the number of refinement levels and the patch size (larger patch sizes are more computational efficient but limit the number of cores and increase memory overhead). Note that `dynamico` usually runs with mass-based vertical coordinates, which adds an additional 15% to its cpu time.

We have also directly estimated the overhead due to multiscale adaptivity for `wavetrisk` by comparing single scale runs (with the largest possible patch sizes for the given resolution for best performance) and multiple scale runs with minimum resolution $J = 4$ and maximum resolutions $J = 6, 7, 8, 9$ on 40 cores with minimum scale $J = 4$. The tolerance was set to zero to give an upper bound on the cost of the multiscale runes. The code solved the Held and Suarez (1994) test case. We found that the overhead due to the multiscale adaptivity is about three times. We therefore conclude that `wavetrisk` is about 4 times slower per active grid point than `dynamico`, and therefore a grid compression factor greater than 4 is required for `wavetrisk` to be faster than `dynamico`. Remapping accounts for about 7% of the cost, so remapping every 10 time steps decreases the overhead difference to about 3.7 times. Using RK33ssp instead of RK4 further reduces the overhead to about 2.8 times. Note that, unlike `dynamico`, we have not made a serious effort to optimize the performance of `wavetrisk` and it is certainly possible to reduce this overhead significantly. Nevertheless, we will show that compression ratios of up to 1000 times are achievable using the adaptive code and it is certainly much faster than the equivalent non-adaptive code for high resolution intermittent problems.

In order to estimate strong parallel scaling performance we ran two Held and Suarez (1994) general circulation experiments. The first run was non-adaptive at horizontal resolution $1/4°$ with 30 vertical levels for ten time steps. This case is balanced and figure 4 shows that it has linear strong scaling from about 8 to at least 2560 cores (the lack of linear scaling for four or few cores is due to the intrinsic overhead of parallel computations).

The second trial was fully adaptive using trend filtering, with a minimum horizontal resolution of $1/2°$ and three levels of refinement to $1/8°$ and 18 vertical levels computed with a relative error tolerance $\varepsilon = 0.08$. This simulation was first run for 500 h to allow the climate dynamics to develop. Figure 5 shows the solution and adaptive grid at t=500 h. The grid compression ratio is a relatively modest 4.5. The simulation was then restarted and run for another 4 h (about 300 time steps) to estimate strong parallel scaling and timing. The computational load was rebalanced on restart, but there was no further rebalancing. The load imbalance (ratio of highest to average load) varied from 3 to 8 during this simulation. This run therefore represents a very unbalanced case. Nevertheless, figure 4 shows reasonable strong scaling up to at least 640 cores. Since weak scaling performance is a better indication of parallel performance than the strong scaling, these results suggest weak scaling should be good to much larger numbers of cores. In fact, the shallow water code on which `wavetrisk` is based showed 70% weak scaling efficiency for as few as 1300 computational elements per core Aechtner et al. (2015). The current three-dimensional code has better parallel performance since the sub-domains are distributed to the cores as complete vertical columns, which mean each core has a larger load.

In summary, `wavetrisk` has a minimum overhead of about 50% compared with `dynamico` associated with the management of the adaptive grids and local geometry. This overhead increases with the number of refinement levels and decreases with patch size. However, the cost per active node is independent of the grid compression ratio. We also show that strong parallel performance is good for $O(10^3)$ cores. This suggests that `wavetrisk` requires a grid compression ratio of at least two to be



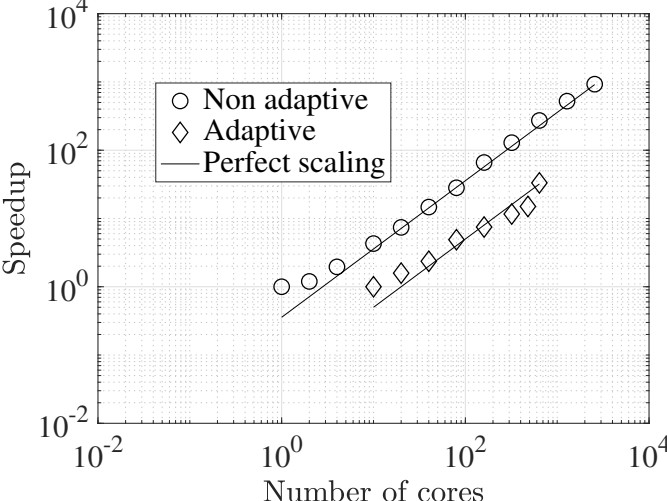

**Figure 4.** Strong scaling of `wavetrisk` on the Compute Canada machine `niagara` for a simulation of the Held and Suarez (1994) general circulation experiment for a perfectly balanced (non-adaptive) run at a resolution of $J = 8$ ($1/4°$) and for a strongly unbalanced (dynamically adaptive) run at a maximum resolution of $J = 9$ ($1/8°$) resolution with trend based error tolerance $\varepsilon = 0.08$.

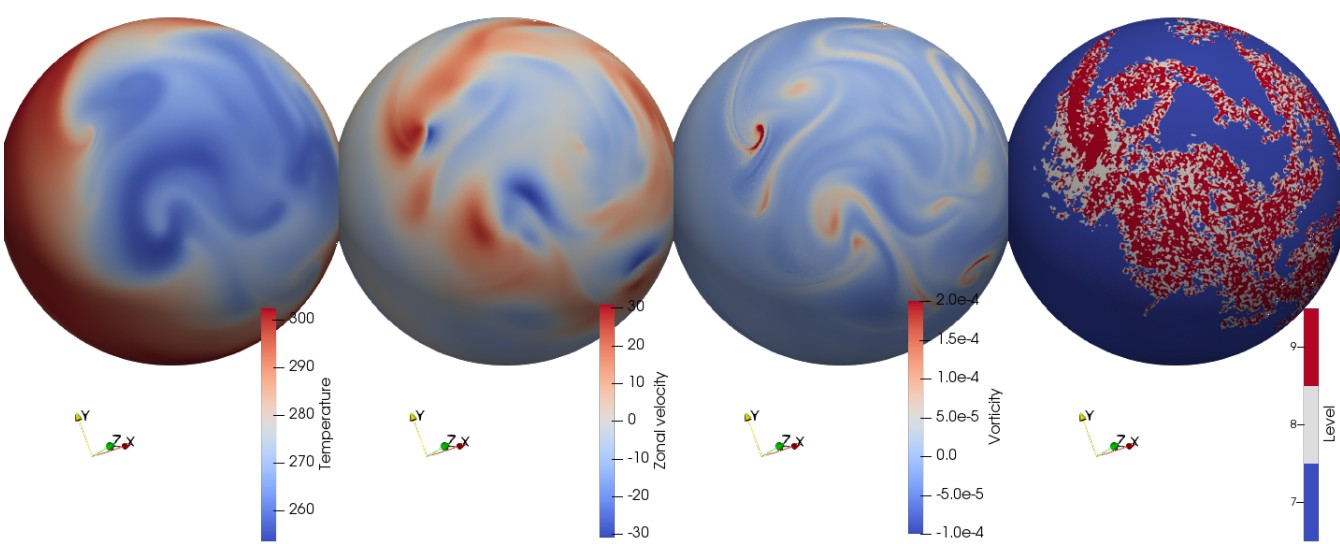

**Figure 5.** Adaptive Held and Suarez (1994) general circulation experiment at t=500 h with maximum resolution $1/8°$ used for the strong parallel scaling results shown in figure 4. Results are shown at the vertical level corresponding to 850 hPa. The rightmost figure is the adaptive grid with a grid compression ratio of about 4.5.



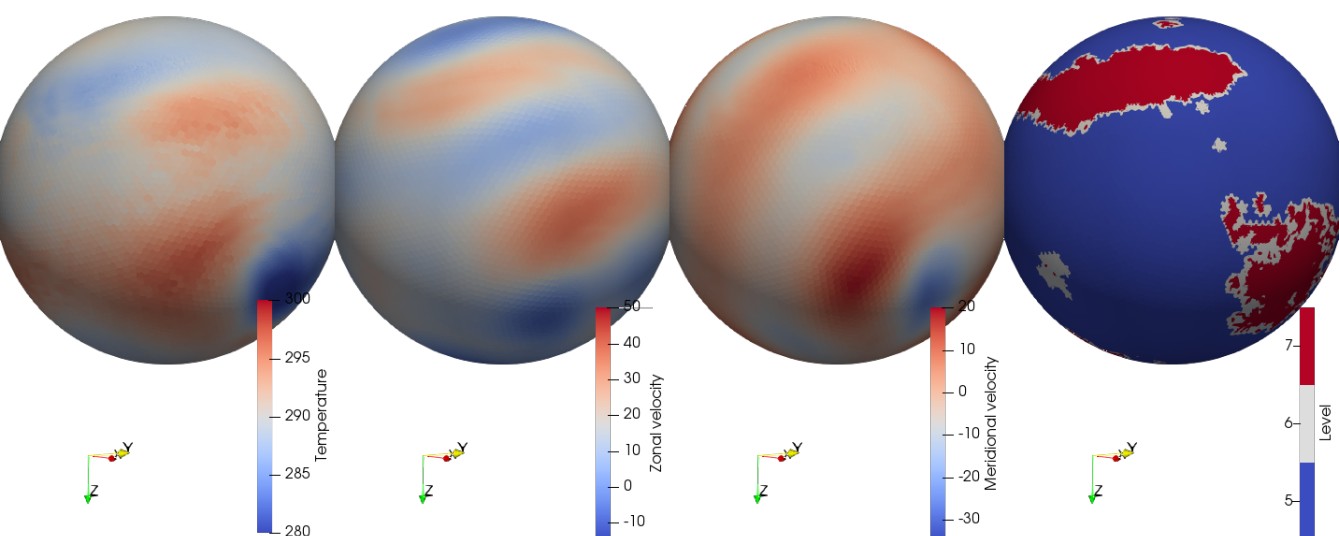

**Figure 6.** Results of the adaptive simulation of the mountain induced Rossby wave case shown at 700 hPa at day 25.

more computationally efficient than an equivalent non-adaptive code. As we show in the following sections, grid compression ratios of 4 to 200 or more are achievable depending on the flow (see figure 11 for example). In general, higher grid compression ratios are achievable at higher resolutions and in more turbulent (intermittent) cases.

### 5.2 DCMIP 2008 test case 5: mountain-induced Rossby wave train

5 The first validation we consider is the mountain-induced Rossby wave train used in the dynamical core model inter-comparison project (DCMIP) in 2008 (Jablonowski et al., 2008). This test case is relatively smooth and does not produce much small scale structure. It is, however, a good test of the ability of the adaptive algorithm to track developing wave instabilities. It is similar to the case described in Tomita and Sato (2004), but with a hydrostatic surface pressure. The simulation starts from smooth, balanced isothermal initial conditions and a Rossby wave train instability is generated by an isolated Gaussian mountain over 10 the course of the first 15 days.

The coarsest scale in the simulation is $J = 5$ (2°) and the finest scale is $J = 7$ (1/2°) with 27 vertical hybrid $\sigma$ pressure levels. The vertical grid is remapped when the thickness of a vertical level has dropped to 30% of its initial value. There is no diffusion. The grid is adapted on the tendency wavelets. We run the simulation for a total of 30 days. The results are shown at day 25 on sphere in figure 6.



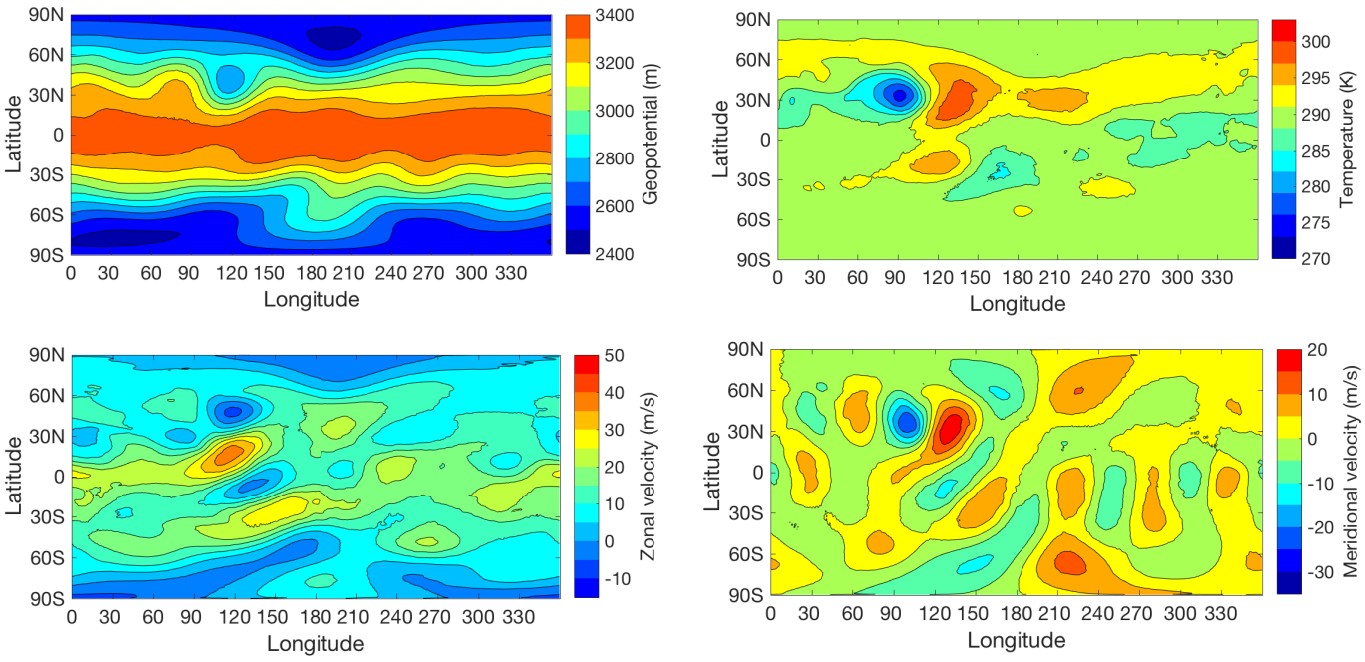

**Figure 7.** Latitude–longitude projections at 700 hPa of the adaptive simulation of the mountain induced Rossby wave test case at day 25.

Figure 7 shows latitude–longitude projections of the results of the adaptive simulation at 700 hPa at day 25. The adaptive data is first interpolated to a uniform $J = 7$ $(1/2°)$ grid, then interpolated to the desired pressure level and finally projected onto the plane. The results are in good qualitative agreement with those shown in the DCMIP 2008 report (Jablonowski et al., 2008). There is some difference in the weaker structures, which is inevitable given that the adaptive simulation necessarily
resolves the more intense structures more highly than the weaker ones.

The mountain induced Rossby wave train is a good first validation of the `wavetrisk`. However, it is relatively smooth, it does not test the ability of the code to deal with intense instabilities and has no "physics" (e.g. cooling, or Rayleigh drag). The following baroclinic instability test case is much more intense and generates small scale vorticity filaments.

### 5.3 DCMIP 2012 test case 4: baroclinic instability of jet stream

Jablonowski and Williamson (2006) proposed a deterministic test case for dry dynamical cores of atmospheric general-circulation models that simulates the evolution of a baroclinic wave in the northern hemisphere. Perturbation of an analytic steady state solution triggers a baroclinic instability that generates a series of vortices. These vortices grow and interact, eventually producing a two-dimensional turbulence-like vorticity field of intense filaments and vortex cores. The rapid development of the vortical instability and its subsequent evolution is a challenging case for an adaptive method. This baroclinic instability
was Test Case 4 of DCMIP 2012.





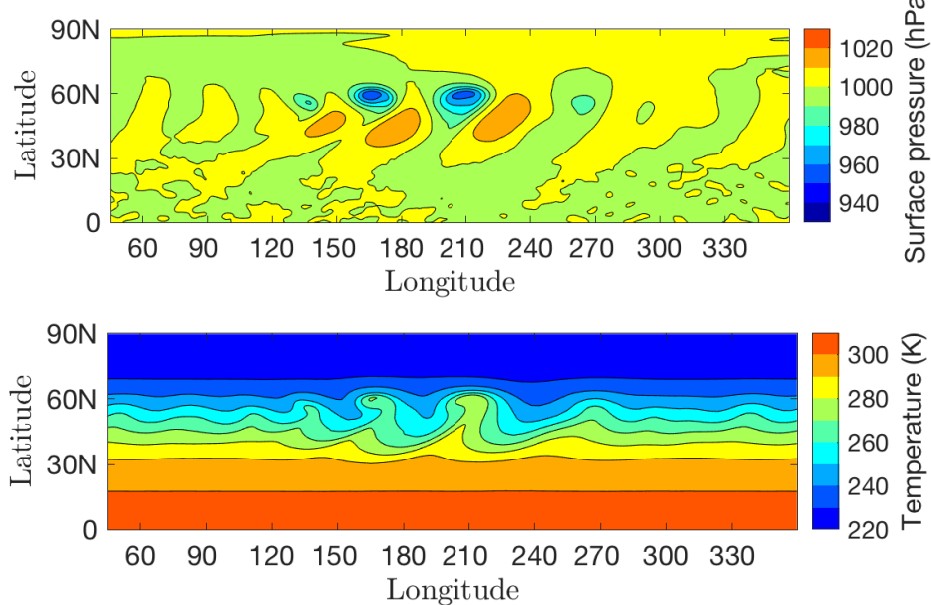

**Figure 8.** Latitude–longitude projections of surface pressure and temperature at vertical level 4 (about 867 hPa) of the adaptive simulation of the baroclinic instability test case at day 9.

With a sufficiently low relative tolerance (e.g. $\varepsilon = 0.03$ when adapting on the solution) `wavetrisk` successfully captures the explosive cyclogenesis around day 8 and subsequent breaking of the wave train around day 12 to produce a series of intense filamentary vortices (see 10).

Figure 8 shows the surface pressure and temperature at day 9, which agree reasonably well with the equivalent results in figure 6 from Jablonowski and Williamson (2006). Note that we do not expect exact agreement since the adaptive simulation does not resolve all regions uniformly. Beyond day 12 the vortices interact to produce turbulence-like flow in both hemispheres. These results therefore validate the ability of the model to capture suddenly developing instabilities and track their evolution to a turbulence-like state.

Next we compare the characteristics of the solution-filtered and trend-filtered variants of `wavetrisk`. In both cases the maximum scale is $J = 7$ ($1/2°$) and we use 27 vertical hybrid $\sigma$ pressure levels as specified in Jablonowski and Williamson (2006). There is no explicit diffusion added to stabilize the dynamics or to damp out grid scale oscillations and both simulations using RK45ssp time integration (Spiteri and Ruuth, 2002).. It is important to note that, although this is a deterministic test case when simulated non-adaptively, adaptivity necessarily neglects some less dynamically important structures and so the details flows with different tolerances eventually diverge once they become turbulent (after roughly 12 days). This is, however, an optimal test case for adaptive methods since at early times only a small portion of the flow is active which allows high compression ratios. In fact, until the instability develops a very coarse grid and large time step is sufficient.



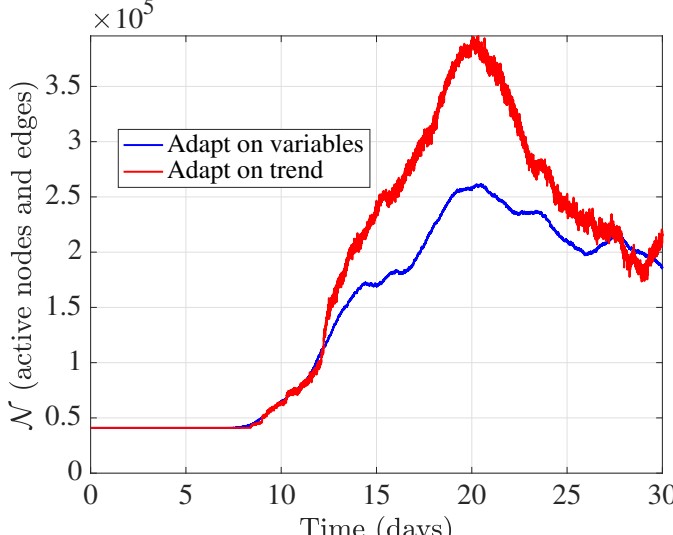

**Figure 9.** Comparison of the $\mathcal{N}$ for the baroclinic instability when adapting on the trend or adapting on the solution. The relative error threshold was set to $\varepsilon = 0.03$ when adapting on the solution and $\varepsilon = 2$ when adapting on the trend. In both cases the active scales are $j = 5, 6, 7$ and there is no additional diffusion. The tolerances were set to achieve similar compression ratios at 9 days. The total number of available grid points is $8.6 \times 10^5$, and so the minimum grid compression ratio is 3.3 when adapting on the solution and 3.3 when adapting on the trend.

Figure 9 compares the grid compression ratios for the baroclinic instability when adapting on the solution and on the trend. The tolerances were set to achieve similar compression ratios at day 9 and were set based on dimensional analysis (i.e. they are fixed in time). Both options use similar numbers of grid points until about day 12 when the flow becomes more turbulent. Once the flow is turbulent adapting on the trend uses significantly more grid points. Although the number of grid points used 5  by both options is similar until day 12, the methods distribute the same number of grid points quite differently.

Figure 10 compares the vorticity fields and active grids at days 9 and 20. At day 9 adapting on the solution distributes the available grid points to track all the developing vortices. In contrast, adapting on the trend concentrates all grid points on the two strongest vortices. In addition, the peak vorticity when adapting on the solution, $3.1 \times 10^{-4}$ s$^{-1}$, is higher than when adapting on the trend, $2.3 \times 10^{-4}$ s$^{-1}$. Once the flow is turbulent, at day 20, adapting on the trend uses about 50% more grid 10  points than adapting on the solution.

Adapting on the solution appears to be more efficient since adapting on the trend is sensitive to grid scale noise in the trend (the solution is smoother than the trend). Adapting on the trend uses fewer overall grid points when the tolerances are rescaled dynamically (i.e. when the trend norms are re-computed at each time step), but the resulting adapted grid is quite sensitive to local fluctuations in the trend. This effect is much less pronounced when diffusion is added. Based on this and other examples 15  we have investigated, adapting on the solution appears to be preferable when there is little or no diffusion added to damp out grid scale noise.





**Figure 10.** Comparison at days 9 and 20 of the grid compression ratios for the baroclinic instability when adapting on the trend or adapting on the solution. The relative error threshold was set to $\varepsilon = 0.03$ (fixed) when adapting on the solution and $\varepsilon = 0.6$ when adapting on the trend. In both cases the resolution is $j = 5, 6, 7$ and there is no additional diffusion. The tolerances were set to achieve similar compression ratios at 9 days. The vorticity field is shown at hybrid vertical level 4 (about 870 hPa).





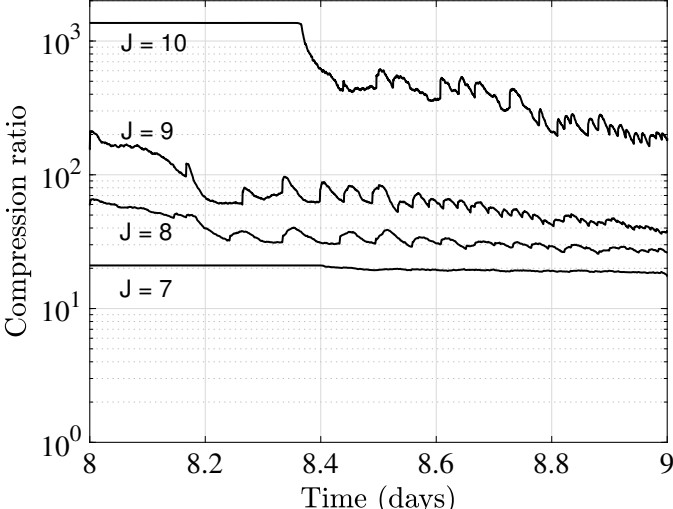

**Figure 11.** Grid compression ratios at the beginning of the baroclinic instability for maximum resolutions $J = 7$ $(1/2°)$, $J = 8$ $(1/4°)$, $J = 9$ $(1/8°)$ and $J = 10$ $(1/16°)$. In all cases the coarsest resolution is $J = 5$ $(2°)$. The grids are adapted on the trend and there is no diffusion. Note that the $J = 10$ uses a higher relative error threshold $\varepsilon = 4$ than the others, which use $\varepsilon = 2$. Compression ratios increase significantly with increasing resolution, reaching as high as 200 times at day 9 for the maximum resolution case. Even at $J = 7$ the code achieves a compression ratio of about 20 at day 9.

It is interesting to investigate how the grid compression ratios change as we increase the maximum allowed resolution. The main advantage of adaptive methods is for intermittent problems that require very high local resolution not attainable using static grid methods. This requires that the grid compression ratio increases significantly with the maximum allowed resolution. Figure 11 compares grid compression ratios for four different maximum resolutions ranging from $J = 7$ $(1/2°)$ to $J = 10$
$(1/16°)$ from day 8 to day 9 at the onset of the instability. The simulations have no diffusion and are adapted on the trends. Unsurprisingly, the compression is very high at the onset of the instability at day 8, over 1300 at $J = 10$. At day 9 once the initial vortices have developed the compression ranges from about 20 at $J = 7$ to about 200 at $J = 10$. Note that we have found that we can also use a higher relative error threshold at higher resolutions which further increases the compression.

This scaling test suggests that the adaptive method should be especially advantageous at high maximum resolutions for
vorticity dominated flows. Although in this flow the instability is highly localized, Aechtner et al. (2015) found compression ratios of about 20 for a homogeneous isotropic shallow water turbulence case on the sphere with scales $J = 5, \ldots, 10$.

### 5.4 Held and Suarez general circulation experiment

Held and Suarez (1994) is a classic test case for dynamical cores of atmospheric general circulation models. It uses simplified "physics" (i.e. radiation and friction/drag models) that nevertheless produce realistic general circulation over relatively short
time scales of $O(10^2)$ days. For example, Liu and Schneider (2010) have used simple Held–Suarez type physics to realistically model the general atmospheric circulation of Saturn. This model uses height-dependent Rayleigh damping to represent





boundary-layer friction and a height- and latitude-dependent Newton cooling relaxation of potential temperature to a prescribed radiative-convective equilibrium. The temperature relaxation includes parameters accounting for cooling at the surface and top of the atmosphere as well as a tropopause. The Held–Suarez general circulation experiment adds a qualitatively new aspect the Rossby wave and baroclinic instability tests we considered above: it includes physics source terms for the temperature and

momentum equations. None of the previous simulations have included these sorts of physics terms, either in three dimensions or in two dimensions on the sphere or on the plane. Nevertheless, we expect the grid adaptation to track the effect of the source terms either directly through their effect on the trend, or indirectly through their effect on the prognostic variables.

We compare low and high resolution runs since previous work has suggested that the choice of coarsest can affect the result. We also want to explore how grid compression changes with resolution (as in the baroclinic test case). The low resolution run

has coarsest scale $J_{\min} = 4$ and finest resolution $J = 6$ (about 120 km or $1°$) with tolerance $\varepsilon = 0.04$. The high resolution run has coarsest scale $J_{\min} = 6$ and finest resolution $J = 8$ (about 30 km or $1/4°$) with tolerance $\varepsilon = 0.02$. The low resolution case is run on 40 cores and the high resolution case is run on 640 cores. As in Wan et al. (2013), both cases start from the Jablonowski and Williamson (2006) zonally symmetric initial condition with random noise of magnitude 1 m/s added to the zonal wind. In both cases the simulations are first run non-adaptively for 200 days at at the coarsest resolution and then

restarted adaptively with the maximum resolution. The time step is adaptive with CFL criterion one. The maximum possible grid compression ratio for both cases is 21.

We use 19 vertical $\sigma$ pressure levels concentrated at the top and bottom of the atmosphere. The vertical grid is remapped using a piecewise parabolic method with WENO limiting every ten time steps.

Small scale noise is damped with $p = 2$ hyperdiffusion with diffusion constant $K_\phi = 3.48 \times 10^{15}, K_\delta = 3.48 \times 10^{16}, K_\omega =$

$2.17 \times 10^{14}$ for the low resolution run and $K_\phi = 2.33 \times 10^{14}, K_\delta = 5.98 \times 10^{14}, K_\omega = 1.46 \times 10^{13}$. The diffusion is applied each time step in the main trend routine. The source terms for the potential temperature and the velocity representing cooling and Rayleigh damping are implemented as a separate Euler step.

Mean and variance statistics are calculated using a parallel version of Welford's inline algorithm (Chan et al., 1983) by first interpolating the solution to the finest grid from checkpoints saved every 24 hours. The second-order statistics are essentially

converged after 200 days (the first order statistics converge much more quickly).

A typical low resolution result is shown in figure 12 (top). The average grid compression ratio at this low resolution is only $1.9 \pm 0.1$ with $\varepsilon = 0.04$. The adaptive algorithm is able to track the development and evolution of the fine scale filamentary vortex structures over long times. Note that since the adapted grid is the union of adapted grids over all vertical levels, the adapted grid does not necessarily correspond exactly to the structures at the vertical level 11 (about 250 hPa) at the level of

the upper atmosphere jets shown in the figure.

Figure 13 shows standard first and second order statistics averaged zonally and over time for the low resolution run. These statistics are qualitatively similar to those of Dubos et al. (2015), Wan et al. (2013) and Lin (2004). (Lin (2004) is the only one that uses Lagrangian vertical coordinates.) The main quantitative difference is in the slightly lower magnitude of the eddy kinetic energy. This difference is due partly to the use of Lagrangian vertical coordinates, where the remapping introduces

additional dissipation, and partly to the additional dissipation generated by the adaptivity at these relatively low resolutions.





Lin (2004) did not show the eddy kinetic energy, but he reported a maximum variance of zonal wind of about 300 m²/s², slightly larger than the 284 m²/s² we find here (not shown), and similar to the 301 m²/s² we find the for the high resolution case discussed below. Note that the original Held and Suarez (1994) paper did not present results for eddy kinetic energy.

The choice of remapping has a significant influence on the eddy kinetic energy. For example, we found that changing from a
piecewise linear remapping to a piecewise parabolic remapping increases the maximum eddy kinetic energy by 53 m²/s² for a non-adaptive simulation with resolution $J = 5$ (240 km). Similarly, Lin (2004) found that including a monotonicity constraint in the remapping lowered the maximum variance of the zonal velocity by about 20 m²/s². We would also like to emphasize that piecewise constant remapping gives qualitatively incorrect results (e.g. zonal jets are too high), presumably because it is too dissipative, and is not a good choice for this test case.

Larger tolerances $\varepsilon$ effectively increase dissipation, which decreases maximum eddy kinetic energy, while choosing a smaller tolerance leads to essentially no compression in the low resolution run. (The other statistics are much less sensitive to $\varepsilon$.) Note that a fixed (non-adaptive) resolution $J_{\min} = 4$ (4°) gives a maximum eddy kinetic energy of only 280 m²/s², much less than the published values of over 400 m²/s² at higher resolutions of 2° and 1°. This suggests that the coarsest grid may be the main factor behind the lower eddy kinetic energy we observe here at higher compression ratios.

In contrast, figure 12 (bottom) shows typical results from the high resolution simulation at $\varepsilon = 0.02$. The grid compression is clearly much higher than in the low resolution run, and the fine scales are limited to a small neighbourhood of the high intensity vorticity filaments. The average grid compression ratio for the high resolution simulation is $7.6 \pm 0.5$, exactly four times as large as in the low resolution case. Both the compression and the fluctuations in the compression are much higher than in the low resolution run. These results suggest that the adaptive method is useful primarily at higher resolutions and for more
turbulent flows. This confirms our earlier observations for two-dimensional shallow water turbulence on the sphere (see figures 17 and 18 of Aechtner et al., 2015). The advantages of the adaptive method should be even greater for maximum resolutions higher than $J = 8$ (1/4°).

Figure 14 shows standard first and second order statistics averaged zonally and over time for the high resolution run. The results are qualitatively very similar to the low resolution run, with the main differences being more intense eddy momentum
and eddy kinetic energy. The negative high altitude zonal jet is also a bit stronger.

The results presented here suggest that the eddy kinetic energy is quite sensitive to the coarsest resolution and relative error tolerance chosen and that, not surprisingly, the adaptive method does not provide significant benefits for low resolution simulations. However, dynamic adaptivity makes much higher resolutions runs feasible at a significantly reduced computational cost (both in terms of memory and cpu time). This advantage should increase for larger maximum resolutions unattainable using
non-adaptive codes. The ability to restart a simulation adaptively at a higher resolution (e.g. add three levels of refinement) allows events of interest to be explored at very high local resolutions, even if it is only for relatively short times.

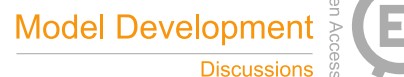

**Figure 12.** Typical results for the low resolution (top) and high resolution (bottom) Held and Suarez general circulation test case at 250 hPa. The grid is adapted on the solution with relative error tolerance $\varepsilon = 0.04$ in the low resolution case and $\varepsilon = 0.02$ in the high resolution case. The grid compression ratio is 2.0 for the low resolution case and 7.5 for the high resolution case.





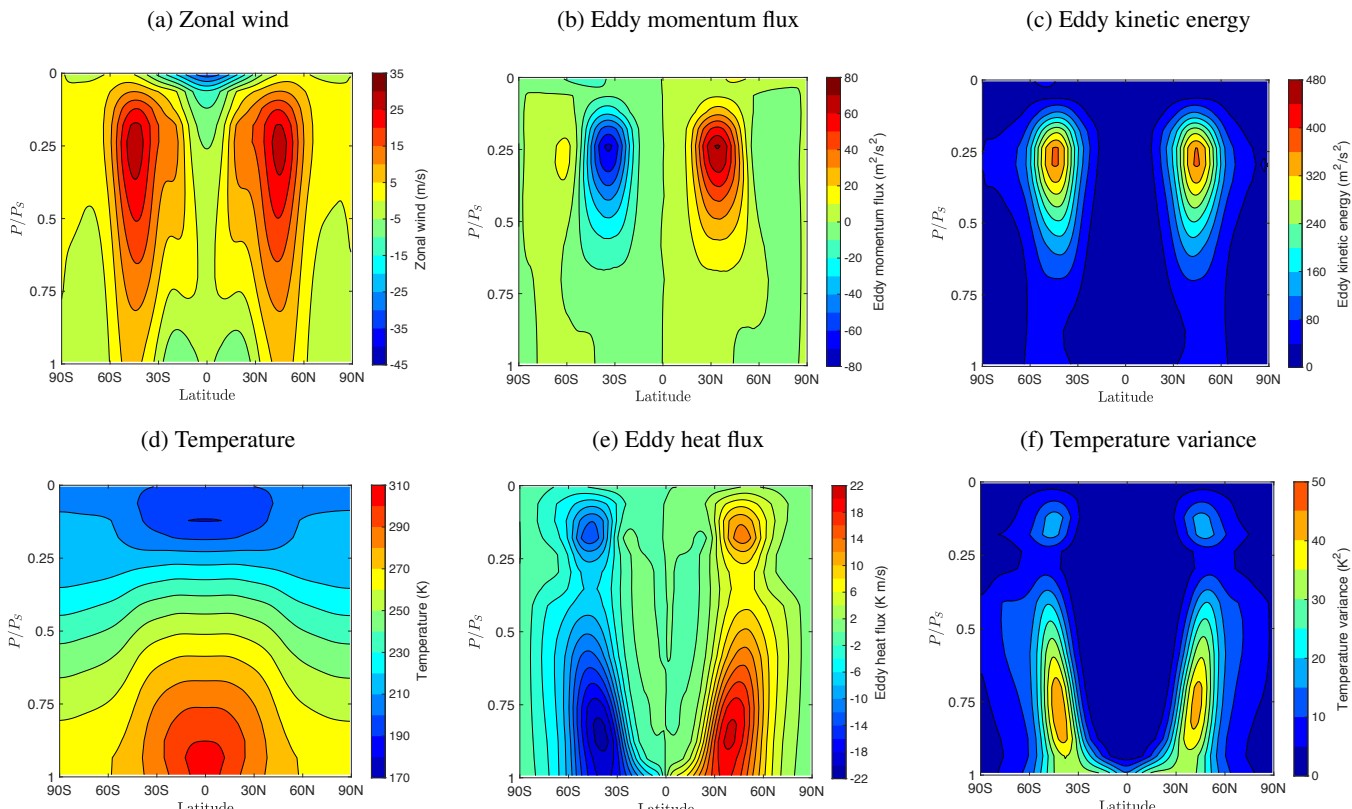

**Figure 13.** Time–zonal statistics for the low resolution Held and Suarez (1994) test case with scales $J = 4, 5, 6$ (maximum resolution $1°$). The grid is adapted on the solution error with tolerance $\varepsilon = 0.04$. Statistics are averaged over 400 days after day 200 by interpolating saved data to the finest grid.

## 6 Conclusions

This paper introduces `wavetrisk`: a new adaptive dynamical core. `wavetrisk` uses the hydrostatic `dynamico` discretization Dubos et al. (2015) with a Lagrangian vertical coordinate. Second-generation discrete wavelet transforms provide control of the relative error of the solution in each vertical layer at each time step. The adaptive grid is uniform vertically and is com-

5 posed of columns of varying horizontal sizes. In addition to the horizontal adaptivity, the vertical coordinates are remapped onto a hybrid $\sigma$-pressure coordinate using an arbitrary Lagrangian Eulerian (ALE) scheme. In principle, ALE allows $r$-adaptivity of the vertical coordinates by optimizing the target grid at each remap. $h$-adaptivity may also be possible in the future by deactivating some vertical layers (the so-called "dormant layers" used in ocean modelling).s

The code is parallelized via domain decomposition using `mpi` and the data is stored in a hybrid quad tree–patch data

10 structure. The computational load is re-balanced at each checkpoint save. We demonstrate excellent strong parallel scaling up to at least 2560 cores in the perfectly balanced case and up to at least 640 cores in an unbalanced case. `wavetrisk` is





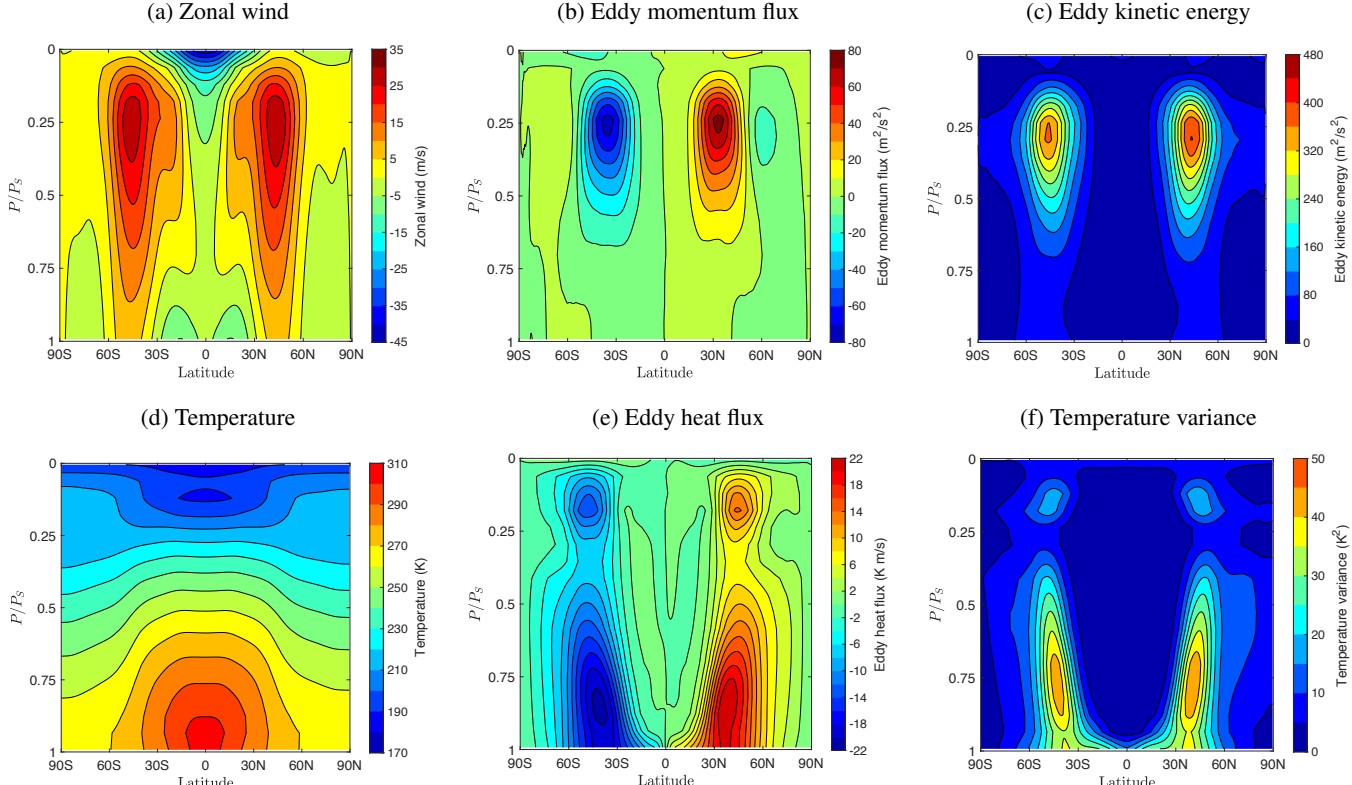

**Figure 14.** Time–zonal statistics for the high resolution Held and Suarez (1994) test case with scales $J = 6, 7, 8$ (maximum resolution $1/4°$). The grid is adapted on the solution error with tolerance $\varepsilon = 0.02$. Statistics are averaged over 200 days after day 700 by interpolating saved data to the finest grid.

three to four times slower per active grid point than the non-adaptive `dynamico` code, which suggests that we require a grid compression ratio of more than four for adaptivity to provide an advantage in cpu time.

The grid may be adapted based on the wavelet coefficients (interpolation errors) of either the prognostic variables themselves (mass density, mass-weighted potential temperature and velocity) or of their associated tendencies. The desired relative error

5  tolerance is multiplied by the appropriate norm (of the variables or tendencies) computed at each time step to control the relative error. Note that the adaptivity is necessarily dissipative compared to an equivalent simulation with uniform resolution on the finest grid since some energy is lost when computational elements are coarsened. However, by construction, this dissipation is controlled and may be reduced by decreasing the tolerance or increasing the maximum allowed resolution. In previous work (Aechtner et al., 2015) we explored how the effective Reynolds number of a turbulent flow depends on the choice of error

10  tolerance parameter.

We have validated the code on three standard benchmarks: a mountain induced Rossby wave train, baroclinic instability of a jet stream and the Held and Suarez general circulation experiment. These tests show that `wavetrisk` correctly captures the





dynamics, including rapidly developing instabilities, with only a small portion of the total grid points available on a similar non-adaptive gird. The grid compression ratio can reach over 200 in ideal cases (e.g. the start of the baroclinic instability with five levels of refinement) and is advantageous at sufficiently high resolutions even in more homogeneous flows like Held and Suarez.

Because adaptive climate simulation is a new field, we have deliberately included many options in our adaptive algorithm. `wavetrisk` can adapt on errors in the solution or on errors in the trend, it can run with no diffusion, Laplacian diffusion, or hyperdiffusion. The vertical grid can be re-gridded (using a large selection of remapping schemes) each time step, or only when a level becomes too narrow. And, of course, we can choose different relative tolerances $\varepsilon$ and maximum and minimum grid resolutions. In many cases, the code is stable without diffusion and with grids adapted either on the solution or the trend.

However, our test cases suggest that adapting on the solution (rather than the trend) generally gives more accurate and faster solutions for a given number of grid points. Including a small amount of diffusion stabilizes the code and reduces the number of active grid points by reducing grid-level noise.

It is clear that the main application of `wavetrisk` is for simulations at maximum resolutions unattainable by non-adaptive dynamical cores. In future work we will use `wavetrisk` to simulate simple Held–Suarez type climates at much higher

resolutions and for longer times and investigate the behaviour of more sophisticated physics parameterizations in adaptive simulations. We are also developing an ocean variant of `wavetrisk` that will improve the ALE formulation of the vertical coordinate and use penalization for bathymetry and coastlines. This work builds on the shallow water ocean model we presented in Kevlahan et al. (2015).

*Code availability.* WAVETRISK-1.0 is published under the Creative Commons 4.0 license at https://doi.org/10.5281/zenodo.2817161.

*Author contributions.* Both NKRK and TD have contributed to the research and paper preparation.

*Competing interests.* The authors have no competing interests.

*Acknowledgements.* NKRK would like to thank NSERC for Discovery Grant funding and Compute Canada for computing time and the Université Grenoble–Alpes and CNRS for visiting professorships. This work was supported by the French national programme LEFE/INSU. The authors would like to thank Matthias Aechtner for his major contributions to the computational foundations of `wavetrisk` which are

outlined in the papers on the shallow water equations version of the code on the sphere.



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
