# Peer review of "WAVETRISK-1.0: an adaptive wavelet hydrostatic dynamical core"

_Geoscientific Model Development, 2019_

## Short Comment (SC1) · 30 May 2019

**1   Comment on the Speed-Performance characterization**

I have read the present manuscript with great interest. I agree with the authors that adaptive methods are likely to become the most effective option for future weather and climate models. As such, the potential for significant scientific impact of this manuscript is large. Unfortunately, the present manuscript does not address the most prominent question that naturally arises when assessing an implementation of adaptive mesh refinement (AMR). I hope to convince the authors to extend the analysis in this work via this comment.

The reason to introduce AMR is related to its speed performance, and the authors make various unsupported claims regarding this aspect. I realize that a complete characterization is impractical, but the authors do not present *any* data that quantify this important detail:

1: The claim in line 19 Page 4 is pivotal and should be proven for `WAVETRISK-1.0`. I like to be shown wrong that $\tau_n$ is a function of the actual grid structure and will vary with the compression ratio for typical atmospheric flows. If there exist a characterizing numbers for $\tau_n$, they should be listed.

2: It does not help that the authors made an effort to non-dimensionalize the vertical axis in fig. 4, instead of listing absolute performance (e.g. $\tau_n$). Updating the figure would allow for simple comparisons between various AMR strategies, and thereby help to distinguish the good from the inefficient approaches for AMR-code development (with the relevant disclaimers). My first suspicion when a code displays near-perfect parallel scaling, is that it is just very slow, such that the overhead of MPI communications is relatively small. I think it is important to show that this is not the case for `WAVETISK-1.0`.

3: The paper does not address the memory requirements for running with high resolutions/compression ratios. Is there adaptive memory management as well?

**2 Furthermore,**

1) The authors make contradicting claims regarding their choices for the refinement indicator. In line 32-33 Page 2, they claim that their formulation for the refinement criterion is clearly defined and objective. Whereas in line 5-6 page 29, they admit to operate in a new field and therefore provide more than one option. I have learned form the authors (et al.), that the multi-resolution analysis is indeed a powerful tool to locally quantify non-trivial polynomial content in the discretized field data. But how it should

be linked to mesh-element-size selection is still an open topic, and it should be clearly presented as such. Further, I think the work of Naddei et al. (2019) and the references therein form a good starting point for the interested reader on this topic.

Naddei, F., de la Llave Plata, M., Couaillier, V., Coquel, F. (2019). *A comparison of refinement indicators for p-adaptive simulations of steady and unsteady flows using discontinuous Galerkin methods.* Journal of Computational Physics, 376, 508-533.

2) The Held-Suarez model has been studied before using an adaptive tree grid by Popinet (2012);
S. Popinet - *Quadtree-adaptive global atmospheric modelling on parallel systems.* Weather and Climate Prediction on Next Generation Supercomputers, Exeter, UK, 22-25 October, 2012.

https://www.newton.ac.uk/files/seminar/20121024100510409-153402.pdf

It makes sense to attribute this work proper.

**3   Finally,**

I did enjoy reading the manuscript and I am convinced this work does represent an important step for the future of atmospheric modeling.

Your sincerely,

Antoon van Hooft

---

## Author Comment (AC1) · 12 Jun 2019

1. (i) The 3D code has exactly the same structure for horizontal adaptivity as the 2D code (there is no vertical adaptivity). It has the same parallelization and same data structure. It therefore inherits the scaling inherits the numerical properties of the 2D code, including the fact that cpu time per grid point is independent of the compression ratio for given coarsest and finest grids. The relevant graph is shown in figure 7(a) of Aechtner et al (2015) cited in the manuscript. Note that this property does not depend on the particular flow considered, or the number of vertical levels (although the actual compression does). For a given compression ratio the cpu time increases proportionally to the number of vertical levels.

(ii) Figure 4 illustrates the parallel scaling, not absolute performance, and so 'speed up' is the right measure (especially since the absolute speed is highly machine dependent). We address the absolute speed of the code compared with the non-adaptive version of the code, DYNAMICO, on pp 27 and 28 where we point out that the wavetrisk is about 3-4 times slower per adaptive grid point than DYNAMICO on the same machine with the same number of cores.

(iii) WAVETRISK-1.0 uses the same hybrid patch-quad tree adaptive memory management structure as the 2D shallow water code described in Aechtner et al (2015), as described on pp 7-8. We have checked that for large numbers of cores the memory used per core is proportional to the number of cores. Memory is therefore not a limiting factor since large problem are run on larger numbers of cores.

As an indication, the total memory used per active grid point is about (30*NZ+2)*8 bytes/1e3 (bytes/kb) = 7.2 kb per grid point for 30 fields on NZ=30 vertical levels. Note that because memory is not a limiting factor, we have not made an effort to minimize memory use per active grid point.

2. (i) Thank you for the reference, we will discuss it in the paper. We think the claims are not contradictory: filtering the wavelet coefficients does provide an objective criterion for grid refinement based on the local polynomial interpolation error of variables and/or trend. (Note that the non-dimensional tolerance is scaled by the relevant norms for each variable.) Apart from refining on the wavelets of the variables or trend, our comments on p 29 refer to other factors determining the accuracy and stability of the method, such as the vertical re-gridding algorithm and use of hyperdiffusion.

(ii) Thank you for letting us know about the unpublished 2012 talk by Popinet et al showing some preliminary results for Held and Suarez, that we were previously unaware of. We will note that his group, as well as the CHOMBO group, have made some (unpublished) significant steps towards developing and evaluating AMR for complex 3D atmospheric flows.

---

## Referee Comment (RC1) · Anonymous Referee #1 · 2 Jul 2019

This manuscript describes the elements used to build a wavelet-adaptive, global, hydrostatic atmospheric model. The numerical scheme is based on a previous, non-adaptive version of the model and is only described briefly in the manuscript. The main ingredients required to make this model wavelet-adaptive are discussed: including important aspects such as conservation of quantities during interpolations between refinement levels. The hexagonal/icosahedral grid structure is presented also in the context of wavelets and general details on the implementation and parallel scheme are given. The algorithm and criteria used for adaptive refinement are presented, including a discussion and analysis of trend- versus solution-based criteria. Validation and performance tests are then conducted, including idealised cases and a more complete, climate-scale Held-Suarez circulation model.

[Figure]

I found the manuscript clearly and carefully written. The results presented are important as they constitute one of the first demonstrations of adaptivity for three-dimensional climate-scale models.

My main reservation regards the amount of details given for the performance of the model. The authors choose to give only relative performance data, either comparing the code with itself or with its previous non-adaptive incarnation, dynamico. Absolute performance should also be given: in particular, the number of integration timesteps, the wall-clock time and some details on the system on which the tests were run (CPU type, memory etc.) must be added for each of the cases.

Further comments on this and other more minor points follow:

* The order of the method is not clearly discussed. I assume it is spatially second-order. Some discussion on possible extensions to higher order should also be included.

* p.2 line 25: "To the best of our knowledge, no previous work has developed and evaluated AMR for complex three-dimensional atmospheric flows." This statement is too broad. There are many earlier references for adaptive three-dimensional atmospheric flows. By "atmospheric", the authors probably mean "global-scale atmospheric" flows. Also, as pointed out in one of the readers' comments, Popinet presented results of an adaptive Held-Suarez model at the "Multiscale Numerics of the Atmosphere and Ocean" Newton Institute program back in 2012.

* p.2 line 30: "However they do not find any "clear strategy for establishing the best general refinement criteria." In contrast, wavetrisk uses objective and clearly defined refinement criteria which control the multiscale relative error of the solution or of its tendencies as measured directly by the wavelet coefficients." Again, here the authors claim too much. The approach presented later in the paper is useful and interesting however it cannot really be said to be a "clear strategy for establishing the best general refinement criteria." Indeed, finding such a strategy is a tall order and has been the topic of numerous publications (and even entire conferences) dealing with "Uncertainty

Quantification".

* Figure 4: Although "speedup curves" are a standard representation, they are a particularly poor way of characterising (parallel) performance. This should be replaced with a figure showing the "speed" or "computational time" per processing unit as a function of the number of processing units. Perfect scaling is then a constant and the value of this constant gives the absolute performance. This thus shows two important values (absolute speed and scalability) instead of one (scalability) and is immune to many of the biases of the "speedup curve" representation.

* p. 3 Section 4 typo: "applies the principle of wavelet-based adaptivity to present the context."

* p.7 line 13. typo "Note that the primal grid of triangles remains nested on the sphere, which means that the restrictions of velocity, Bernoulli and circulation and straightforward."

* p. 8 line 22: "To remedy this we use a simple rebalancing algorithm to redistribute sub-domains amongst the cores to produce a more balanced load. This rebalancing is done at each checkpoint save." This is an extremely short description of a non-trivial and important algorithm. More details should be given and/or appropriate references given.

* Algorithm 1, typo: "at all vertical levels so final adapted grid is union of adapted grids over all vertical levels."

* p.16 line 7, typo: "on the cost of the multiscale runes"

* p. 24 line 3. typo: "The Held–Suarez general circulation experiment adds a qualitatively new aspect the Rossby wave and baroclinic instability tests we considered above:"

---

## Referee Comment (RC2) · Anonymous Referee #2 · 28 Aug 2019

Review of the GMDD manuscript:

WAVETRISK-1.0: an adaptive wavelet hydrostatic dynamical core

Authors: Nicholas K.-R. Kevlahan and Thomas Dubos

**Summary:**

The manuscript documents the design of a new wavelet-based, dry, hydrostatic 3D atmospheric dynamical core on the sphere with embedded 2D (horizontal) adaptive mesh refinement (AMR). The 3D model is built upon the 2D AMR shallow water model described in Aechtner et al. (2015). The 3D extensions of the model equations and the numerical treatment of the vertical dimension are based on the work by Dubos et al. (2015). The manuscript briefly reviews the conservation properties and numerical design of the model, the hexagonal/icosahedral grid structure, the wavelet-based AMR approach, the parallelization technique, data structure, and the AMR selection criterion. The latter is based on a numerical error tolerance  $\varepsilon$ , and not on commonly-used flow features (like vorticity) of the simulation. The dynamical core and its AMR & parallel performance characteristics are tested via standard test cases which include a deterministic mountain-induced Rossby wave train, a baroclinic instability test case and the climate-like Held-Suarez test.

Overall, the manuscript is very well written. It is an important contribution to the literature since atmospheric 3D AMR research on the sphere is still extremely sparse. My comments below mostly suggest minor changes and clarifications. However, the quality of most spherical figures (their color schemes) should be improved to more clearly visualize the flow fields.

- 1) Abstract: Most journals do not allow references in the abstract. Check the GMD guidelines.
- 2) Reformulate page 2, line 25. It is true that adaptive 3D atmospheric flows, especially in spherical geometry, have not been extensively studied in the past. There are, however, at least two Ph.D. theses in the literature (Jablonowski (2004) and Ferguson (2018)) and have some discussion of 3D dynamical cores on the sphere with dynamic grid refinements. There are more examples for 3D atmospheric flows in limited-area AMR models, like e.g. Skamarock and Klemp (Mon. Wea. Rev., 1993).
- 3) Page 9, line 12, also page 27, line 3: Typo, either state 'by Dubos et al. (2015)' or use citation format '(Dubos et al., 2015)
- 4) Page 12, line 13 and 25: The AMR criterion needs to be clarified. Are the grid adaptations invoked if all three indicators  $\varepsilon_{\mu}$ ,  $\varepsilon_{\theta}$ ,  $\varepsilon_{u}$  exceed the threshold or is it enough that one of them exceeds the tolerance? It is not clear how the maximum norm is computed. Please provide some insight into its evaluation in this wavelet-based method.
- 5) Page `5, line 28, define the acronym DCMIP
- 6) Figures 5, 6, 10, 12: Provide more explanation for the selected map projection. Are these stereographic projections? Which point are they centered on (e.g. north-polar stereographic projections)? The blue-red color scheme for all flow figures is very hard to read. Please improve the quality of the figures and e.g. use the examples by Aechtner et al. (2015), Figs. 17 and 18.
- 7) Figs. 9 and 10: It it intentional that Fig. 9 documents the parallel characteristics with  $\varepsilon$ =2 (when adapting on the trend) and  $\varepsilon$ =0.06 in the actual visualization of the flow? The tolerances also differ when the adaptations on the solution are documented. This these two

figures both document the baroclinic wave test case, why is the reader not shown the flow fields t(Fig. 10) hat correspond to Fig. 9? Explain or modify.
8) Page 24, line 19-20: Add the physical units for the diffusion coefficients
9) Page 25, line 2: typo, should read '... we find for the ...'

---

## Author Comment (AC2) · 24 Sep 2019

Thank you for your helpful comments, especially regarding improvements to the reporting of parallel performance. Please find our detailed responses below.

Significant changes to the manuscript (other than typo corrections) appear in BLUE in the revised version.

COMMENT: My main reservation regards the amount of details given for the performance of the model. The authors choose to give only relative performance data, either comparing the code with itself or with its previous non-adaptive incarnation, dynamico. Absolute performance should also be given: in particular, the number of integration timesteps, the wall-clock time and some details on the system on which the tests were

run (CPU type, memory etc.) must be added for each of the cases.

RESPONSE: We have added detailed performance information in the new table 2.

COMMENT: The order of the method is not clearly discussed. I assume it is spatially second-order. Some discussion on possible extensions to higher order should also be included.

RESPONSE: As mentioned, the model is based on DYNAMICO, which is based on the TRiSK scheme (which is second order). We now mention this on on page 3. We also mention that our adaptive method could be used any of flux-based scheme, indendent of order. Developing such higher order schemes is outside the scope of the paper, which focuses on making adaptive an existing scheme.

COMMENT: p.2 line 25: "To the best of our knowledge, no previous work has developed and evaluated AMR for complex three-dimensional atmospheric flows." This statement is too broad. There are many earlier references for adaptive three-dimensional atmospheric flows. By "atmospheric", the authors probably mean "global-scale atmospheric" flows. Also, as pointed out in one of the readers' comments, Popinet presented results of an adaptive Held-Suarez model at the "Multiscale Numerics of the Atmosphere and Ocean" Newton Institute program back in 2012.

RESPONSE: On page 2 we cite more previous work developing 3D atmospheric AMR codes. However, no other global 3D AMR method has appeared in a peer-reviewed publication. This is an important distinction since it is difficult to evaluate the robustness and state of development of codes that have been reported only in unpublished conference presentations or theses. For example, Ferguson's thesis notes explicitly that "Several stability problems were observed in both test cases that need to be better understood and corrected." and that the grid refinement criteria require more investigation (his code is based on the general purpose CHOMBO AMR library).

Similarly, the results reported in the conference publication by Popinet (cited in another

comment) show obvious errors in the variance of the potential temperature for the Held-Suarez. It is possible that these issues can be easily resolved, but there is still some development work left to do. (We have discovered that with adaptive codes there is a big gap between "almost works" and "definitely works"...)

COMMENT: p.2 line 30: "However they do not find any "clear strategy for establishing the best general refinement criteria." In contrast, wavetrisk uses objective and clearly defined refinement criteria which control the multiscale relative error of the solution or of its tendencies as measured directly by the wavelet coefficients." Again, here the authors claim too much. The approach presented later in the paper is useful and interesting however it cannot really be said to be a "clear strategy for establishing the best general refinement criteria." Indeed, finding such a strategy is a tall order and has been the topic of numerous publications (and even entire conferences) dealing with "Uncertainty Quantification".

RESPONSE: On page 3 we have added additional text to make clear that we are not claiming that our approach has completely resolved the problem of error control.

COMMENT: Figure 4: Although "speedup curves" are a standard representation, they are a particularly poor way of characterising (parallel) performance. This should be replaced with a figure showing the "speed" or "computational time" per processing unit as a function of the number of processing units. Perfect scaling is then a constant and the value of this constant gives the absolute performance. This thus shows two important values (absolute speed and scalability) instead of one (scalability) and is immune to many of the biases of the "speedup curve" representation.

RESPONSE: We have added a figure with this more informative way of characterizing parallel scaling, along with accompanying text. However, we have decided to retain the the speed-up plot since it is nevertheless a standar representation and so allows for comparison with other methods.

COMMENT: p. 3 Section 4 typo: "applies the principle of wavelet-based adaptivity to

present the context."

RESPONSE: Corrected.

COMMENT: p.7 line 13. typo "Note that the primal grid of triangles remains nested on the sphere, which means that the restrictions of velocity, Bernoulli and circulation and straightforward."

RESPONSE: Corrected.

COMMENT: p. 8 line 22: "To remedy this we use a simple rebalancing algorithm to redistribute sub-domains amongst the cores to produce a more balanced load. This rebalancing is done at each checkpoint save." This is an extremely short description of a non-trivial and important algorithm. More details should be given and/or appropriate references given.

RESPONSE: We have added more details about the rebalancing algorithm on page 9.

COMMENT: Algorithm 1, typo: "at all vertical levels so final adapted grid is union of adapted grids over all vertical levels."

RESPONSE: Corrected.

COMMENT: p.16 line 7, typo: "on the cost of the multiscale runes"

RESPONSE: Corrected.

COMMENT: p. 24 line 3. typo: "The Held–Suarez general circulation experiment adds a qualitatively new aspect the Rossby wave and baroclinic instability tests we considered above:"

RESPONSE: Corrected.

---

## Author Comment (AC3) · 24 Sep 2019

Thank you for your helpful suggestions for improving the clarity of the paper. Please find our detailed responses below.

Significant changes to the manuscript (other than typo corrections) appear in BLUE in the revised version.

COMMENT: 1) Abstract: Most journals do not allow references in the abstract. Check the GMD guidelines.

RESPONSE: Thanks for pointing this out. The guidelines state "Reference citations should not be included in this section, unless urgently required," I guess these references are not really "urgently required", and so we have removed them.

COMMENT: 2) Reformulate page 2, line 25. It is true that adaptive 3D atmospheric flows, especially in spherical geometry, have not been extensively studied in the past. There are, however, at least two Ph.D. theses in the literature (Jablonowski (2004) and Ferguson (2018)) and have some discussion of 3D dynamical cores on the sphere with dynamic grid refinements. There are more examples for 3D atmospheric flows in limited-area AMR models, like e.g. Skamarock and Klemp (Mon. Wea. Rev., 1993).

RESPONSE: On page 2 we now cite this previous work developing 3D atmospheric AMR codes. However, no other global 3D AMR method has appeared in a peer-reviewed publication. This is an important distinction since it is difficult to evaluate the robustness, accuracy and state of development of codes that have been reported only in unpublished conference presentations or theses. For example, Ferguson's thesis notes explicitly that "Several stability problems were observed in both test cases that need to be better understood and corrected." and that the grid refinement criteria require more investigation (his code is based on the general purpose CHOMBO AMR library).

Similarly, the results reported in the conference publication by Popinet (cited in another comment) have obvious errors in the variance of the potential temperature for the Held-Suarez test case. It is possible that these issues can be easily resolved, but there is still some development work left to do. (We have discovered that with adaptive codes there is a big gap between "almost works" and "definitely works"...)

COMMENT: 3) Page 9, line 12, also page 27, line 3: Typo, either state 'by Dubos et al. (2015)' or use citation format '(Dubos et al., 2015)

RESPONSE: I think you may have mis-read the sentence on p 9: the citation starts a new sentence:

Dubos and Kevlahan (2013)} found that good weak parallel efficiency is possible with as few as 1300 computational elements per core in adaptive runs. The three-dimensional code has better parallel efficiency because the column structure of the data produces

a higher computational load for each active grid element.

We have corrected the citation form on p 27.

COMMENT: 4) Page 12, line 13 and 25: The AMR criterion needs to be clarified. Are the grid adaptations invoked if all three indicators $e_\mu$, eq, eu exceed the threshold or is it enough that one of them exceeds the tolerance? It is not clear how the maximum norm is computed. Please provide some insight into its evaluation in this wavelet-based method.

RESPONSE: We have clarified on p 12 that a node is labelled active if any associated scalar wavelet (i.e. mass or mass-weighted potential temperature wavelets) is active at any vertical layer, and an edge is labelled active if its associated vector wavelet (i.e. velocity wavelet) is active in any vertical layer. The maximum norm for each vertical layer is calculated in the usual way: the maximum absolute value of the relevant variable over all active grid points and all scales.

COMMENT: 5) Page '5, line 28, define the acronym DCMIP

RESPONSE: Done.

COMMENT: 6) Figures 5, 6, 10, 12: Provide more explanation for the selected map projection. Are these stereographic projections? Which point are they centered on (e.g. north-polar stereographic projections)? The blue-red color scheme for all flow figures is very hard to read. Please improve the quality of the figures and e.g. use the examples by Aechtner et al. (2015), Figs. 17 and 18.

RESPONSE: All plane projections are simple equidistant cylindrical map projections on a rectangular latitude-longitude grid, which is standard for many test cases (e.g. Galewesky et al (Tellus 2004) and DCMIP 2008) to avoid complications of different map projections. This is now mentioned in the text on p 18 line 8.

We have re-done all spherical figures using a clearer paraview colour palette similar to that used in the previous paper.

COMMENT: 7) Figs. 9 and 10: Is it intentional that Fig. 9 documents the parallel characteristics with e=2 (when adapting on the trend) and e=0.06 in the actual visualization of the flow? The tolerances also differ when the adaptations on the solution are documented. This these two figures both document the baroclinic wave test case, why is the reader not shown the flow fields (Fig. 10) that correspond to Fig. 9? Explain or modify.

RESPONSE: We meant to show the results with the same thresholds. We have corrected this inconsistency and have taken this opportunity to re-run the simulations with the more accurate quadratic PPR remapping. (Because PPR is less diffusive than the original piecewise constant scheme, we used slightly larger thresholds to have a similar number of active grid points at 9 days.) Note that the results are qualitatively similar to the previous version.

COMMENT: 8) Page 24, line 19-20: Add the physical units for the diffusion coefficients

RESPONSE: Done.

COMMENT: 9) Page 25, line 2: typo, should read '. . . we find for the . . .'

RESPONSE: Done.

---

## Author Response (AR2)

**Responses to Referee 1**

Thank you for your comments on the computational performance of WAVETRISK. Please find our detailed responses below. Significant changes are indicated in blue in the revised manuscript.

**Comment**

*1) Since the computational cost can be assumed to be proportional to the number of levels, it should be reported as cpu/(columns\*level/core) rather than cpu/(columns/core). This would allow direct comparison between benchmarks using a different number of levels.*

**Response**

Figure 4 (right) has been changed to use this measure (and we now use active nodes rather than active degrees of freedom, i.e. nodes and edges).

**Comment**

*2) Table 2 is useful but should also include the cost i.e. an additional column with something like*
*Rossby wave 0.86\*160/1.94e5 = 0.7 ms /27 0.026 ms*
*Baroclinic instability 0.40\*40/1.64e5 = 0.1 ms /27 0.0037 ms*
*Baroclinic instability 1.13\*40/4.07e5 = 0.1 ms /27 0.0037 ms*
*Held-Suarez 1 degree 0.27\*40/1.11e5 = 0.1 ms /19 0.005 ms*
*Held?Suarez (1/4 ) 0.29\*320/4.54e5 = 0.2 ms /19 0.01 ms*

**Response**

We have added this information to Table 2 (and use active number of nodes, rather than active number of degrees of freedom).

However, we would like to emphasize that this measure does not take into account speed up due to adaptivity, and hence does not really measure the true computational efficiency of the adaptive code. It is useful only to estimate how well the code has been optimized and compare its overhead to non-adaptive codes. We explicitly addressed overhead (and optimisation) in Section 5.1, where we compared the performance of WAVETRISK with DYNAMICO, which uses the same basic algorithm (although its computational implementation is quite different). We estimate the overhead as about 50% in the non-adaptive case and about 4 times in the adaptive case.

Although an optimized and computationally efficient code is obviously important, we developed WAVETRISK to demonstrate the possibilities of dynamical adaptivity to reduce computational cost, increase accuracy and allow extremely high local resolutions compared with a similar non-adaptive model (DYNAMICO, in this case). Figure 11 shows how adaptivity can accelerate codes at high resolutions: compression ratios of 200 and higher are achievable.

**Comment**

*3) The data in the Table above does not seem consistent with the data reported in Figure 4, right, with values around 1 ms, rather than 0.1 ms here. This must be checked/clarified. Similarly, why is the Rossby wave case so "expensive"?*

**Response**

We have modified Figure 4 (right) by dividing by the number of vertical levels. The times are slower than reported for Held-Suarez in Table 2 because we adapted on the trend (instead of variables), the code was compiled using gfortran (instead of ifort) and we used RK45ssp time integration instead of RK4 (which requires an additional trend evaluation). Please note that when we did the scaling study our intent was to measure parallel speed up, not absolute speed, which is why we did not use precisely the same set-up as the later test case runs at lower resolution. We explain this difference in the caption to Figure 4.

There was a typo in the cpu/dt value (it should have been 0.4 s). The caption to Table 2 now explains that the Rossby wave case (besides adapting on the tendency, which is more expensive) uses a 4x4 patch size, rather than 8x8, in order to run on 160 cores with Jmin=5. The trade-offs involved in the choice of patch size and number of domains at the coarsest scale (and hence the maximum number of cores) are explained in section 5.1.

**Comment**

*4) p. 25 "the high resolution case is run on 640 cores", yet Table 2 says 320 cores. Which is it?*

**Response**

It is 320 cores (640 cores was the maximum used for the adaptive parallel scaling runs using the Held-Suarez test case). This typo has been corrected.

**Comment**

*5) To avoid any ambiguity "milliseconds" should be spelled out somewhere.*

**Response**

This is spelled out in the caption to Figure 4.

**Comment**

*6) The most favourable case of 0.0037 ms (per timestep per degree of freedom) does not compare well at all with other implementations of comparable numerical schemes. For example for the hydrostatic, sigma-coordinate, ROMS code, this cost was reported to be 2e-5 ms (almost 200 times faster!) and this back in 2005/2007... (I didn't look for more recent numbers) see e.g.*

*Zuo, Yue, Xingfu Wu, and Valerie Taylor. "Performance Analysis and Optimization of the Regional Ocean Model System on TeraGrid." TeraGrid'07 Conference.*

*Wang, Ping, et al. "Parallel computation of the regional ocean modeling system." The International Journal of High Performance Computing Applications 19.4 (2005): 375-385.*

*Even though ROMS has been highly optimised, one would expect even naive implementations of explicit shallow-water solvers to have absolute performance of order at least 1e-4 ms (per timestep per degree of freedom, /dtdf).*

*I sincerely hope that this is due to a trivial error in the numbers above. If this is not, then this must be discussed in detail (i.e. possible explanations and solutions should be proposed) and the conclusions on the usefulness/applicability of the method/implementation should be toned down (a*

| Cores | Performance |
|------:|-------------|
| 256 | 0.1141 ms |
| 128 | 0.0410 ms |
| 64 | 0.0180 ms |
| 32 | 0.0074 ms |
| 16 | 0.0049 ms |

Table 1: ROMS performance figures from Lupo et al 2017.

*lot). Note that the numbers given above for ROMS are indicative only (more recent numbers would most probably be even less favourable) and the authors should look for performance data for other (atmospheric) solvers.*

**Response**

The performance figures you cite from these papers **clearly cannot be correct**, and are more than two orders of magnitude faster than ROMS's actual performance. Guillaume Roullet (LOPS, University of Brest) who is a user and developer of ROMS, has confirmed that, depending on the configuration, a cost of a few microseconds is normal for ROMS, and less than 1e-3 ms is out of reach on current supercomputers. In addition, the figures you cite would imply that ROMS is up to 800 times faster than CAM3 (using results from Jablonowski and Williamson 2006 cited in the manuscript)!

Before making such extreme claims it would have been prudent to double check them against other results.

Contrary to your claim, "more recent numbers" confirm that the figures from the two papers you cite are clearly incorrect. The more recent paper Lupo et al (Enhancing Regional Ocean Modeling Simulation Performance with the Xeon Phi Architecture, Ocean 2017, IEEE) "presents a detailed exploration of the acceleration of the Regional Ocean Model System (ROMS) software with the latest Intel Xeon Phi x200 architectures". It reports well-documented results (for the upwelling test case), comparable to WAVETRISK and atmospheric codes, shown in Table 1. It is not credible that the 2007 version of ROMS running on slower machines would be more at least 240 times faster than the 2017 version running on the latest processors!

Note that, because it is a regional model, ROMS can use a regular Cartesian grid, which is more efficient than the irregular grids (such as WAVETRISK's icosahedral grid) that are required on the sphere.

Jablonowski and Williamson (QJR Meteorol Soc 2006, 132, pp 2943–2975) present performance results for three different atmosphere models: NCAR CAM3 (finite volume), NCAR CAM3 (spectral Eulerian) and the GME icosahedral model from the German Weather office (DWD) applied to the baroclinic instability test case we consider in the paper. These performance results are from the same period as the two papers you cite, and should therefore be directly comparable in terms of code development and computer performance. The equivalent performance measures are summarized below (32 Power PC cores) are shown in Table 2.

WAVETRISK performs similarly to these highly optimized operational atmosphere models,

| Model | Performance |
|---|---|
| GME | 0.0057 ms |
| CAM3 FV | 0.0039 ms |
| CAM3 Eulerian | 0.0157 ms |
| WAVETRISK | 0.0173 ms (160 cores) |

Table 2: Performance figures for the baroclinic wave test from (Jablonowski and Williamson 2006).

even without taking into account the speed-up due to adaptivity. Because WAVETRISK achieves a compression ratio of 5.2 for this test case, its true "adaptive" performance is 0.0035 ms. Greater compression ratios (i.e. at higher resolutions with more scales) improve WAVETRISK's performance. Note that an exact comparison requires running the codes on the same machine and the same numbers of cores.

Recall that we actually did such an apples-to-apples comparison of WAVETRISK with (non-adaptive) DYNAMICO, and found a minimum overhead of approximately 50% in the non-adaptive case and an overhead of about four times in the adaptive case (see Section 5.1). Therefore, WAVETRISK should be faster than DYNAMICO if the compression ratio is greater than 4. This overhead is likely similar compared with other non-adaptive "research" codes.

Finally, Heinzeller et al (Geosci. Model Dev., 9, 77–110, 2016) present recent results for the MPAS climate model. The results are slower than the above results: 0.039 ms (120 km grid on 160 cores) to 0.16 ms (3 km grid on 32768 cores). The slower speed is not surprising since MPAS is a full Earth systems model (atmosphere/ocean/land-ice). But even so, it not possible that MPAS running on the fastest current computers would be up to 1000 times slower than ROMS running on two 15 year old machines over a transcontinental ethernet connection!

**Comment**

*p.23 "Nevertheless, figure 4 (left) shows reasonably linear speed up to at least 640 cores."*

*"The more sensitive measure of strong scaling in figure 4 (right) is further from a constant (with a maximum variation of 3 times), although there is not a definite trend with increasing numbers of cores."*

*The scaling for the adaptive case in figure 4, left or right, is definitely not linear. Figure 4-right is not "a more sensitive measure" than figure 4-left (it is exactly the same data), it just leaves less room for wishful interpretation. It is obvious that the two sets of points on figure 4-left have different slopes. The non-adaptive case indeed shows linear scaling (from 8 cores up), as also confirmed more clearly on figure 4-right, while the adaptive case shows a roughly power-law scaling with an exponent clearly lower than 1 (maybe even 1/2? it would be good to indicate this), as more clearly shown on figure 4-right.*

**Response**

A linear fit to the log-log plots shows a scaling of 0.78 and we now indicate this in the caption to Figure 4 and in the text.

We have changed "more sensitive" to "more precise".

**Comment**

*p.28 "We demonstrate excellent strong parallel scaling up to at least 2560 cores in the perfectly balanced case and up to at least 640 cores in an unbalanced case." The second part of this sentence needs to be modified to reflect the previous comment.*

**Response**

The text has been revised.

[revised manuscript text omitted]